

# A novel nature-inspired feature selection algorithm for efficient moisture estimation in fruits using RF-Sensed data

Ali Roman[1], Youssef Altherwy[2], Syed Rameez Naqvi[1,3] and Anas Alsuhaibani[2]

[1] Department of Computer Engineering, COMSATS University Islamabad, Wah Campus, Wah, Punjab, Pakistan
[2] Department of Information Systems, College of Computer Engineering, Prince Sattam Bin Abdulaziz University, Al-Kharj, Saudi Arabia
[3] Department of Computer Science, Tulane University, New Orleans, New Orleans, LA, United States

## ABSTRACT

We propose a novel hybrid nature-inspired feature selection algorithm that unifies update mechanisms from Grey Wolf Optimizer (GWO), Artificial Bee Colony (ABC), and Bat Algorithm (BA). The resulting framework enables optimized machine learning models for precise grape moisture estimation from radio frequency (RF)-sensed data, addressing key challenges in smart agriculture. Performance is assessed in two phases: (1) pairing the feature selection method with a gated recurrent unit (GRU) model and comparing it against benchmark optimizers, and (2) integrating it with a customized convolutional neural network variant (CNN-R) designed for regression. The proposed feature selection technique demonstrates superior performance across all evaluation metrics. When combined with GRU, it achieves significantly lower root mean square error (RMSE) and mean absolute error (MAE) alongside a higher $R^2$ (best-case: 0.999) compared to benchmark methods. With CNN-R, it maintains equally competitive results, validating its architecture-independent effectiveness. Crucially, the study shows that convolutional neural network (CNN), when adapted through CNN-R, can rival traditional regression models like GRU on numerical data.

## INTRODUCTION

Precision agriculture utilizes radio frequency (RF) waves for measuring various aspects of plants, soil and environment, thus making RF-sensing a crucial component. By leveraging the latter, data on crop health, water consumption, stress levels, product and soil moisture content, insect infestation, and other critical crop health indicators may be acquired in (near) real-time. Consequently, farmers can optimize crop harvesting, pesticide use, fertilization, and irrigation (*Bouri, Arslan & Şahin, 2023*; *Oliveira et al., 2024*).

Moisture estimation in fruits is crucial for various agricultural and industrial applications. Accurate moisture levels can significantly influence quality control, storage,

Corresponding author
Youssef Altherwy,
y.altherwy@psau.edu.sa

and processing decisions. Traditionally, RF-sensing is considered non-destructive, and a reliable method for capturing moisture content (*Altherwy & McCann, 2020*). However, the challenge lies in effectively analyzing this data to extract meaningful insights (*Mohyuddin et al., 2024*).

While RF-sensing allows farmers to monitor a range of environmental parameters.The use of machine learning frameworks can help the farmers in improving crop yields and resource efficiency. Farmers can get information on crop health, the best times to plant, watering needs and fertilization schedules (*Mathi, Akshaya & Sreejith, 2023*). The existing methodologies often rely on machine learning models such as recurrent neural networks (RNN) for moisture estimation, coupled with nature-inspired (NI) optimization algorithms for feature selection. While these methods have shown promise, they often fall short in terms of accuracy and computational efficiency. Current research has yet to fully explore the potential of advanced deep learning architectures and enhanced optimization techniques.

It is well known that the performance of a machine learning process, in terms of both accuracy and computational complexity, is largely dependent upon feature extraction and subsequent selection. While the former is typically done by training a suitable deep learning architecture, numerous optimization algorithms have recently been developed for efficient feature selection. The NI optimization algorithms are currently considered state-of-the-art for features selection (*Kwakye et al., 2024*). However, despite the abundance of such methods, there is no one-for-all solution due to the diversity of the dataset and the nature of the application. As a result, researchers have to look for the most suitable options for their target application, thereby creating a room for further exploration and innovation.

The aim of this study is to address the aforementioned limitations by developing a novel framework that leverages a customized convolutional neural networks (CNN) architecture, specifically tailored for regression-based tasks, since they inherently suit the classification problems in image-based applications (*Kattenborn et al., 2021*). Furthermore, we propose a mutation-enhanced hybrid optimization algorithm, which takes inspiration from collaborative hunting behavior of the Grey Wolf Optimizer (GWO) algorithm (*Mirjalili, Mirjalili & Lewis, 2014*; *Mishra & Goel, 2024*). The primary difference between the proposed and the original GWO algorithms becomes visible when two of the best three, $\alpha$, $\beta$ and $\delta$, wolves (representing the best solutions in an iteration), are replaced by bees and bat from the Artificial Bee Colony (ABC) (*Abu-Mouti & El-Hawary, 2012*; *Katipoğlu, Mohammadi & Keblouti, 2024*) and Bat Algorithm (BA) (*Yang & Hossein Gandomi, 2012*; *Jamei et al., 2024*) respectively. This update allows the proposed mechanism to leverage foraging and echolocation principles from the two algorithms, respectively. In order to avoid the problem of being stuck in the local minima, further diversity is introduced by incorporating a mutation of the genetic algorithm (GA) (*Mahmud et al., 2024*). We will demonstrate that the proposed optimization algorithm improves feature selection and overall model performance in the sections to come.

Our approach is divided into two main parts. First, we benchmark various nature-inspired optimization algorithms with gated recurrent unit (GRU)

(*Dey & Salem, 2017*; *Akilan & Baalamurugan, 2024*) model, and evaluate their performance using root mean square error (RMSE), mean absolute error (MAE), and $R^2$ metrics. The reason for selecting the GRU model stems from our previous work in which the GRU model outperformed long short-term memory (LSTM) (*Yu et al., 2019*), bidirectional LSTM (BI-LSTM) (*Huang, Xu & Yu, 2015*), GoogleNet (*Szegedy et al., 2015*) and RESNET-50 (*He et al., 2016*) models for the similar application on the same dataset (*Altherwy et al., 2024*). In the second part, we introduce the proposed optimization algorithm, and pair it with GRU and our customized convolutional neural network variant (CNN-R) model in turn, and demonstrate significant improvement in fruits' moisture estimation accuracy. Thus, the main contributions of this work are summarized as follows:

1. We develop a customized CNN architecture for regression tasks in moisture estimation.
2. We propose a mutation-enhanced hybrid optimization algorithm named Hybrid Predator Algorithm (HPA), for superior features selection.
3. We empirically validate the proposed methods, showing improved performance compared to the existing approaches.

The rest of the article is organized as follows: Background and related work are discussed in 'Related Work'. 'Methods', summarizes the materials and methods used in this work, which include the dataset and algorithms. The proposed framework is presented in 'Proposed NI Algorithm and CNN-R'. Simulation results and statistical analysis are given in 'Results', before we conclude the study in 'Conclusion'.

# RELATED WORK

RF-sensing offers a resilient, non-destructive, and cost-effective solution for precision agriculture by enabling continuous crop monitoring under diverse environmental conditions. Its ability to capture rich plant and soil data without physical contact makes it vital for early disease detection and growth analysis. However, challenges such as signal variability, environmental noise, and high-dimensional data demand advanced machine learning techniques to fully exploit RF-sensing capabilities. In the first part of this section, we outline the principles of RF-sensing, its broad application potential, and its integration with modern machine learning methods. In the second part, we review recent studies highlighting the role of machine learning in advancing smart agriculture.

## RF-sensing: principles and applications

RF-sensing is a technology that uses environmental RF signal detection to interpret certain physical occurrences (*Lubna et al., 2022*). It functions according to the theory of analyzing radio frequency electromagnetic waves. Although it has been used for many years for a variety of purposes, such as wireless communication, radars, and security systems, the development of artificial intelligence (AI) and the internet-of-things (IoT) has revolutionized RF-sensing technology (*Bolisetti et al., 2017*). Among the many contemporary uses of RF-sensing are occupancy, motion, and gesture detection—all of which can be done without the use of cameras. RF-sensing technology has other beneficial uses in agriculture, such as estimating the moisture content of different fruits, and

**Table 1 Related work regarding RF-sensed data.**

| Reference | Year | Research idea | Implementation |
|---|---|---|---|
| *Oliveira et al. (2024)* | 2024 | Grass quality estimation | Random Forest |
| *Abuhoureyah, Wong & Mohd Isira (2024)* | 2024 | WiFi-based human activity recognition | RNN, LSTM |
| *Khan et al. (2023)* | 2023 | Human activity recognition | CNN, LSTM, Hybrid |
| *Zhang et al. (2023)* | 2023 | Grain mass estimation | Multiple variable linear regression |
| *Altherwy & McCann (2020)*, *Altherwy (2022)* | 2022 | Grape moisture estimation | Regression model |
| *Hao et al. (2022)* | 2022 | Safe and dangerous driving classification | Bi-LSTM, GRU |
| *Hameed et al. (2022)* | 2022 | RF-based lip reading framework | VGG16 |

noninvasive diagnostics of physiological parameters in healthcare, such as blood glucose monitoring. We provide a brief summary of some helpful uses for this technology in Table 1.

Table 1 summarizes various research works related to RF-sensed data, detailing their publication year, research idea, and implementation techniques. Key points include: grass quality estimation by *Oliveira et al. (2024)*, which utilized a random forest algorithm to assess the quality of grass based on RF signals (*Oliveira et al., 2024*). *Abuhoureyah, Wong & Mohd Isira (2024)* developed a WiFi-based human activity recognition system using RNN and LSTM networks to analyze activity patterns. *Khan et al. (2023)* focused on human activity recognition, employing a combination of CNN, LSTM, and Hybrid models to improve accuracy and efficiency. *Zhang et al. (2023)* estimated grain mass using multiple variable linear regression, demonstrating the potential for RF-based agricultural applications. *Altherwy & McCann (2020)*, *Altherwy (2022)* estimated grape moisture content with a regression model, showcasing an innovative approach to agricultural monitoring. *Hao et al. (2022)* differentiated between safe and dangerous driving behaviors using Bi-LSTM and GRU models, highlighting advancements in automotive safety. *Hameed et al. (2022)* implemented an RF-based lip reading framework with the VGG16 architecture, illustrating the use of CNNs in advanced communication systems. The table highlights the diversity of applications and machine learning techniques employed in smart agriculture.

## Selected deep learning models' applications

The RNNs (*Yu et al., 2019*) and their more complex variations, LSTM, Bi-LSTM, and GRU, are crucial for sequence-based applications and are also good for estimation. Using memory cells and gating methods, LSTM networks preserve long-term relationships in order to circumvent the vanishing gradient issue that limits conventional RNNs (*Sherstinsky, 2020*). By gathering context from previous and future states and processing data in both forward and backward directions, Bi-LSTM networks improve this capability (*Huang, Xu & Yu, 2015*). With the forget and input gates combined into a single update gate, GRUs are a less complex alternative to LSTMs that are computationally economical, which frequently achieve equivalent performance (*Dey & Salem, 2017*). While Bi-LSTMs

are superior in total context capture, GRUs are superior in computational efficiency, and LSTMs are superior in long-sequence learning.

CNNs are especially good at processing images because of their capacity to automatically learn the spatial hierarchies of features through convolutional layers (*Brahmi, Jdey & Drira, 2024*). CNNs were first created for classification problems, but they have also been effectively modified for regression-based applications and remote sensing applications (*Kattenborn et al., 2021*). CNNs predict continuous values instead of discrete classifications in various applications. Let us go through some examples of these deep neural networks in smart agriculture to highlight their significance and impact.

Recent advancements in smart farming and precision agriculture have leveraged deep learning and optimization techniques to enhance prediction accuracy and operational efficiency. A particle swarm optimization (PSO)-CNN-Bi-LSTM model—a hybrid optimization-enabled deep learning approach—for smart farming applications is introduced in *Saini & Nagpal (2024)*. A novel model combining Shuffled Shepherd Optimization with attention based convolution neural network with optimized bidirectional long short term memory (ACNN-OBDLSTM) to predict brinjal crop yield, showcasing significant improvements in yield estimation is proposed in *Rao et al. (2024)*. An IoT-based prediction and classification framework utilizing adaptive multi-scale deep networks, offering robust solutions for various smart farming challenges is developed in *Padmavathi et al. (2024)*. To optimize water usage, an intelligent irrigation scheduling using a deep bidirectional LSTM technique is proposed which support sustainable agriculture practices (*Jenitha & Rajesh, 2024*). An automated weather forecasting and field monitoring system using a gated recurrent unit (GRU)-CNN model integrated with IoT, aiming to enhance precision agriculture through accurate environmental monitoring is proposed by *Akilan & Baalamurugan (2024)*. Lastly, to address the critical issue of intrusion detection in IoT-based smart farming, a hybrid deep learning framework, ensuring secure and resilient farming operations is developed by *Kethineni & Pradeepini (2024)*.

Transformer-based architectures and, more recently, Mamba-based state-space models have been developed as a result of recent developments in deep sequence modeling. For example, SenseMamba (*Huang et al., 2025b*) combines Kolmogorov-Arnold Networks (KANs) with state-space modeling to provide wireless human sensing in real-time and with minimal overhead for a variety of activities. By incorporating dilated convolutions and Mamba-inspired elements for reliable temporal feature extraction, BiMamba Kolmogorov Arnold based Transformers (BiMKANsDformer) (*Huang et al., 2025a*). improves on conventional transformers for water quality prediction in environmental applications. Similar to this, RSMamba (*Zhao et al., 2024*) show how state-space-inspired structures allow for precise and computationally effective analysis in tasks involving remote sensing and visual tracking, respectively. These developments demonstrate the increasing applicability of transformer and Mamba-based models in tasks involving sequential, high-dimensional, or dynamic data, which encourages their use in plant disease modeling and smart agriculture.

## NI feature selection

In machine learning, feature selection is the process of locating and choosing a subset of pertinent characteristics from a dataset in order to enhance model performance (*Theng & Bhoyar, 2024*). A number of advantages, including increased accuracy, decreased overfitting, quicker training times, and greater interpretability, results from this approach. Models can improve their ability to adapt to new environments, generate more accurate predictions, and be simpler to understand by concentrating on the most important elements (*Chandrashekar & Sahin, 2014*).

Techniques for NI feature selection usually prove more effective by mimicking the natural processes. These methods search for the best answers by simulating physical, biological, or social systems. Examples include the genetic algorithm (GA) (*Mahmud et al., 2024*), which emulates the process of natural selection, Particle Swarm Optimization (PSO) (*Marini & Walczak, 2015*), which is based on bird flocking behavior, Ant Colony Optimization (ACO) (*Dorigo, Birattari & Stutzle, 2006*), which is based on ant foraging. The Artificial Bee Colony (ABC) (*Abu-Mouti & El-Hawary, 2012*), which is modeled after honey bee foraging; Bat Algorithm (BA) (*Yang & Hossein Gandomi, 2012*), which is based on bat echolocation. The Grey Wolf Optimization (GWO) (*Mirjalili, Mirjalili & Lewis, 2014*), is based on the social hierarchy and hunting behavior of grey wolves, and Red Fox Optimization (RFO) (*Połap & Woźniak, 2021*), which is based on red fox hunting method. When examining vast and intricate feature regions, these techniques work very well.

By utilizing global search capabilities, adaptability, resilience, and flexibility, the NI strategies improve machine learning performance (*Singh et al., 2024*). These algorithms have an inherent tendency to avoid getting stuck in local optima, identify (near) optimal solutions, and conduct a thorough search in the feature space. Their robustness and flexibility enable hybridization with other methods, resulting in more effective, accurate, and interpretable models (*Abu Khurma et al., 2022*). Their adaptation to complicated contexts makes them appropriate for high-dimensional datasets.

Recent advancements in optimization algorithms have led to the development of various hybrid techniques aimed at enhancing performance in complex problem-solving scenarios. *Dhal & Azad (2024)* proposed a hybrid momentum accelerated Bat Algorithm combined with GWO for efficient spam classification, showcasing significant improvements in accuracy and speed. *Umar et al. (2024)* introduced a modified Bat Algorithm designed to tackle complex and real-world problems with enhanced solution quality and robustness. *Lakshmiramana et al. (2024)* presented a hybrid approach integrating ACO and ABC techniques for optimal resource allocation in cognitive radio networks, demonstrating superior resource management.

In the domain of autonomous systems, *Ketafa & Al-Darraji (2024)* developed the RFO-GWO Optimization Algorithm for path planning in autonomous mobile robots, ensuring efficient and collision-free navigation. *Águila-León et al. (2024)* focused on optimizing photovoltaic systems using a GWO-Enhanced PSO algorithm to improve maximum power point tracking (MPPT) controllers, resulting in enhanced energy efficiency and system performance. *Hu et al. (2024)* applied a GA-GWO Hybrid Algorithm for scheduling container transportation vehicles in surface coal mines, achieving optimized

scheduling and increased operational efficiency. These studies highlight the versatility and effectiveness of hybrid NI optimization techniques in addressing a wide range of challenges across different domains.

## METHODS

This section provides insights to the data acquisition, machine learning methods and several benchmark nature-inspired feature selection algorithms we employ for estimating grapes moisture. Our primary focus will be on GRU as our previous study suggests better performance in-terms of RMSE, MAE and $R^2$ (*Altherwy et al., 2024*). This is the primary reason of using feature selection techniques with GRU, but we will also see the impact of our proposed HPA technique with LSTM and Bi-LSTM. We specifically focus on two machine learning methods namely GRU and CNN, both of which have proven extremely successful in estimation and classification tasks using numeric and imagery data respectively. Following up, in 'Results', we demonstrate that our proposed novel feature selection algorithm HPA, when combined with either machine learning model, manages to outperform the benchmark methods in terms of estimation accuracy.

### Comparison with previous work

This section briefly compares the current work with one of our previous works done regarding grapes moisture estimation (*Altherwy et al., 2024*). Figure 1 presents the comparison. The figure shows a concise P2 package from *Altherwy et al. (2024)*. The enhancements within the existing framework are also shown along P2 package. The first major enhancement is in feature selection, where the techniques are increased form three (PSO, ACO and GA) to eight including our main contribution, the HPA. Secondly, in our previous work, we came to the conclusion of using GRU as our ML model after a detailed performance analysis. In the current work, we retain the GRU model for our analysis, but also introduce a customized CNN-R model tailored for the regression tasks. While this study is independently focused on regression-based moisture estimation in fruits, it builds on our broader exploration of nature-inspired feature selection in smart agriculture. A prior work by the authors (*Ali et al., 2025*) addressed a different multiclass classification on image based agricultural datasets, using a lighter variant of the feature selection algorithm and a CNN with a softmax and classification head.

### Dataset: acquisition and description

The dataset employed in this research is based on the experimental component of the work presented in *Altherwy (2022)*, *Altherwy & McCann (2020)*. The data is collected using a wireless sensing system from real grape clusters. Both experimental and simulation setups were originally developed, with simulations conducted using CST Microwave Studio. However, for this study, only experimental (real) cluster data were used.

In the experimental setup, there were nine real grape clusters. Each cluster was suspended from a wooden rod and mechanically rotated to capture signal responses at 30 different angles. Two directional antennas were aligned, operating at a frequency band of 5 to 6 GHz. For each cluster, four separate files were generated. These files correspond to the

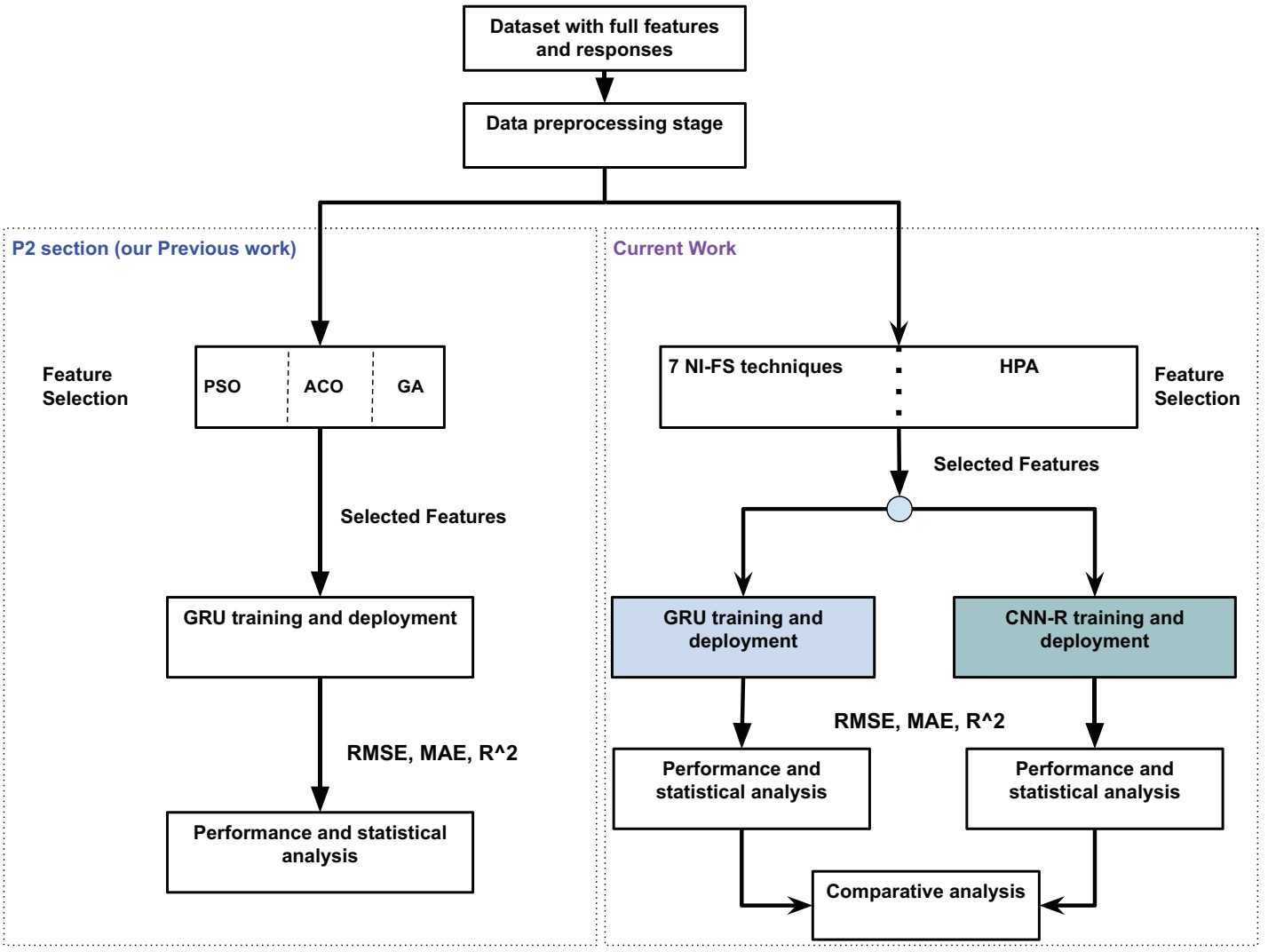

**Figure 1 Comparison with our previous work (grape moisture estimation).**

scattering parameters S11, S12, S21, and S22, respectively. Each file records 1,601 frequency points for each of the 30 angles. For each frequency-angle combination, five signal features are available: the raw complex value, dB strength, magnitude, and phase in both radians and degrees.

The moisture and sugar content of each grape cluster are summarized in Table 2.

Although the original dataset includes both experimental and simulated data, this study exclusively uses the experimental measurements for model development and evaluation. To facilitate data exploration, the dataset is publicly available at (*Roman et al., 2025*).

## Working with LSTM, Bi-LSTM and GRU

The choice of LSTM, Bi-LSTM, and GRU was motivated by the fact that the dataset discussed in 'Dataset: Acquisition and Description' is sequential in nature. RNN with LSTM architectures are used to identify long-term dependencies in sequential data. They

| Table 2 Moisture/sugar content of the grape clusters. | | | | | | | | | | |
|---|---|---|---|---|---|---|---|---|---|---|
| **Cluster type** | **Content** | | | | | | | | | |
| Real | Sugar | 82.84 | 53.17 | 38.08 | 22.28 | 88.1 | 69.82 | 53.14 | 37.63 | 26.5 |
| | Moisture | 380 | 233 | 157 | 95 | 369 | 282 | 215 | 150 | 103 |
| Simulated | Sugar | 12 | 20 | 48 | 21.7 | 47 | 80 | 25 | 33 | 70 |
| | Moisture | 55 | 98 | 120 | 92 | 132 | 190 | 125 | 160 | 200 |

accomplish this by adding memory cells and gating mechanisms (forget, input, and output gates) that control the information flow. These gates let LSTM models learn from lengthy sequences and mitigate the vanishing gradient issue that plagues conventional RNN by enabling them to maintain and update a cell state over time. Bi-LSTM networks analyze data in both forward and backward directions, increasing the capabilities of ordinary LSTM. Because Bi-LSTM models take into account both past and future information in the sequence, this dual processing enables them to have a thorough comprehension of the context. This is especially helpful for jobs like natural language processing, where performance can be enhanced by context from both sides.

Another variant of RNN architecture that makes the design of LSTM simpler is the GRU network. They blend the cell state and hidden state together and combine the input and forget gates into a single update gate. GRU models are a popular option when computational resources are constrained because of their simplified structure, which usually lowers computational complexity. With our dataset in particular, they performed well as compared to LSTM and Bi-LSTM counterparts.

## Working with CNN

CNN are a class of deep learning algorithms primarily used for image and video recognition tasks. They work by automatically learning spatial hierarchies of features through backpropagation. CNN consists of multiple layers, each with a specific role:

1. **Convolutional layers:** These layers apply a set of filters (or kernels) to the input image. Each filter slides over the input, performing a convolution operation that captures local features such as edges, textures, and patterns. The result is a set of feature maps, which highlight the presence of specific features in the input image.
2. **Normalization and activation layers:** After the convolution operation, batch normalization is performed if required. The activation function is typically the rectified linear unit (ReLU) function, which is applied to introduce non-linearity into the model. This allows the CNN to learn complex patterns.
3. **Pooling layers and drop out layers:** These layers perform downsampling operations, such as max pooling or average pooling, to reduce the spatial dimensions of the feature maps. Pooling helps to reduce the computational load, memory usage, and the number of parameters in the network, while retaining the most important information.
   The dropout layer in a CNN is a regularization technique used to prevent overfitting, improve generalization, and enhance the network's robustness.

4. **Fully connected layers:** Towards the end of the network, fully connected layers, similar to those in traditional neural networks, are used to integrate features extracted by the convolutional and pooling layers. These layers contribute to the final classification or regression tasks by mapping the high-level features into output categories.

5. **Output layer:** The final layer of a CNN is typically a softmax layer for classification tasks, providing probabilities for each class label, or a linear layer for regression tasks, providing continuous output values.

## Benchmark NI feature selection techniques

In this research, we employ a variety of NI optimization techniques for feature selection, leveraging the inherent strengths of each algorithm to achieve robust and efficient selection. The GA is known for its ability to explore a vast search space through crossover and mutation, making it suitable for identifying optimal feature subsets. ACO mimics the foraging behavior of ants and excels in discovering high-quality solutions through pheromone trails, making it effective for feature selection in complex datasets. PSO, inspired by the social behavior of birds, efficiently navigates the search space by considering both individual and collective experiences, ensuring rapid convergence to optimal feature sets. The ABC algorithm utilizes the foraging behavior of bees to balance exploration and exploitation, resulting in a comprehensive feature selection process. The RFO algorithm, based on the hunting strategy of red foxes, introduces a unique perspective on optimization by emphasizing strategic moves and adaptability. The BA leverages echolocation to dynamically adjust the search process, making it highly adaptable to varying feature landscapes. Lastly, the GWO emulates the leadership hierarchy and hunting mechanism of grey wolves, offering a balanced exploration and exploitation strategy. By incorporating these diverse and NI optimization techniques, we ensure a thorough and multifaceted approach to feature selection, enhancing the overall performance and robustness of the model.

While the proposed NI algorithm takes inspiration from a subset of these benchmark algorithms, we carry out its thorough comparison with each of them. The latter–based on estimation accuracy measured in terms of RMSE, MAE and $R^2$–is detailed in 'Results'.

# PROPOSED NI ALGORITHM AND CNN-R

This section begins with the description of the proposed NI algorithm for feature selection. We discuss its origin, present its through mathematical model followed by highlighting its convergence conditions and complexity analysis for a fair comparison with the benchmark algorithms. The second part of the section discusses how we tailor the CNN to suit a regression-based task. We name the resulting model CNN-R.

## Proposed NI feature selection

### Inspiration

The proposed NI feature selection algorithm named onwards as Hybrid Predator Algorithm (HPA), takes its inspiration from the GWO algorithm. In order to ensure comprehensive exploration while concentrating on favorable regions, GWO constantly

modifies its search method using a leadership hierarchy. Faster convergence and increased efficiency are facilitated by its straightforward parameter setting, which minimizes the need for tedious adjustment. Furthermore, GWO's resilient and broadly applicable methodology allows it to efficiently manage a wide range of data distributions, yielding better performance measures including reduced RMSE, MAE, and increased $R^2$ values. Courtesy to our exhaustive simulations, we conclude that its hierarchical search strategy enables the GWO algorithm to align well with the layered structure of the CNN, and, hence, outperform its counterparts in effective feature selection.

On the other hand, the GRU networks demand fine-tuning of temporal dependencies, for which exploration of the GWO approach does not prove effective. This usually leads to suboptimal performance in capturing temporal dependencies. In such situations, BA and ABC algorithms prove more effective with their dynamically adjusting search strategies, which helps in optimizing the recurrent connections in the GRU networks (*Ahmad et al., 2024*).

### HPA mainframe

The mainframe of the HPA is structured on the GWO's template. The latter is a meta-heuristic algorithm, in which the best three solutions represent the alpha, beta and delta wolves leading the pack in a hunt. Initialization, exploration, and exploitation are the three main phases that make up the GWO algorithm.

The initial grey wolf population is created randomly within the search space during the initialization phase. Every wolf stands for a potential fix for the optimization issue. These wolves' locations are represented in the solution space as vectors. The exploration phase is similar to how wolves scout their surroundings in order to find prey. Wolves positions are updated in accordance with the relative positions of alpha, beta, and delta wolves. During the exploitation phase, the wolves systematically converge toward the best solutions identified thus far. The alpha, beta, and delta wolves take the lead in this process, employing a refined, greedy strategy to enhance the search for the optimal solution. By continuously updating their positions to be closer to the leading wolves, the pack effectively narrows the search space, allowing for a more precise and focused optimization of the candidate solutions.

The alpha wolf signifies the best solution discovered up to that point, guiding the overall direction of the search process. The beta wolf, representing the second-best solution, assists the alpha in steering the search, offering additional guidance to maintain diversity. The delta wolf, as the third-best solution, supports both the alpha and beta wolves, helping to prevent premature convergence by exploring other promising regions of the search space. The position update mechanism involves calculating the new positions based on the weighted influence of the alpha, beta, and delta wolves. The final position update for a wolf is the average of the influences from these three leaders. *Ahmad et al. (2024)*, succinctly summarize the GWO framework and points out its strengths.

### Adaptation

The HPA is a merger of three NI-FS techniques, GWO, ABC and BA, along with mutation process adopted from the GA. As stated earlier, the mainframe of the proposed scheme is

the same as of the GWO algorithm, whereas, the other algorithms are integrated as the position update components within. This way, while the GWO component ensures suitability with the CNN model, the other algorithms take the lead when coupled with the GRU model. These algorithms were chosen for hybridization due to their complementing skills in navigating high-dimensional search environments and their ability to successfully strike a balance between exploration and exploitation.

GWO offers a strong search mechanism that directs the search toward favorable areas of the solution space by using the alpha, beta, and delta wolves, which stand in for the best solutions in the population. Effective exploration and exploitation are ensured by this hierarchical structure, which makes GWO especially well-suited for optimization tasks involving static feature selection.

The dynamic search capabilities of ABC and BA, which modify the search method in real-time, led to their incorporation. GWO is effective at selecting static features, but it is not always able to capture dynamic temporal dependencies, which are crucial for recurrent models such as GRU networks. Adaptive search algorithms are used by ABC and BA to overcome this constraint, enabling HPA to effectively handle intricate, non-linear correlations in the data.

Distinct from conventional metaheuristic feature selection techniques, which are often designed for classification or static data modeling, HPA is specifically designed for regression tasks with nonlinear, dynamic dependencies. This hybridization strategy combines structured exploration (GWO), dynamic foraging (ABC), and velocity-based adaptation (BAT) and is novel in the context of regression based tasks and allows for superior convergence and robustness.

Through the hybridization of these algorithms, HPA is able to use the adaptive search capabilities of ABC and BA for more dynamic tasks while simultaneously leveraging GWO's stable exploration for feature selection. Improved feature selection performance is the outcome of this synergy for a variety of models and datasets, including RF-sensed data.

Upon confirming these attributes through preliminary simulations, we arrive at a unique hybrid of these algorithms that enjoys the best of both worlds—discussed next.

Iterative optimization is the foundation of HPA's feature selection procedure. Each wolf's location serves as a candidate solution in the form of a feature subset at the center of this procedure. By using the chosen features to train a regression model (such as linear regression) and computing the RMSE between the predicted and actual values, the fitness of each solution is assessed. Because the wolf locations are adjusted to reduce the RMSE, this procedure guarantees that only the most pertinent features are chosen.

GWO's hierarchical leadership structure, in which the alpha, beta, and delta wolves direct the solution space's exploration and exploitation, serves as the basis for the position updates. Furthermore, the search dynamics brought about by ABC and BA guarantee that the algorithm maintains its quest for the globally optimal feature set without being stuck in local optima.

HPA can effectively handle high-dimensional feature spaces by utilizing this hybrid optimization technique, which makes it possible to computationally choose the best feature set for RF-sensed data. The method works especially well for regression tasks where

accuracy and resilience are crucial since it can adjust to both static and dynamic dependence.

The HPA algorithm uses several techniques that are included into its architecture to alleviate the curse of dimensionality. Initially, HPA uses the advantages of ABC for dynamic candidate solution refinement, BAT for adaptive local search, and GWO for structured exploration during the feature selection process. By methodically removing superfluous or irrelevant features, these complementing techniques lower the complexity of the dataset without sacrificing its quality.

Second, HPA uses a fitness function designed to maximize regression performance metrics like RMSE, as seen by the goal function's use of linear regression (fitlm). Each feature subset is assessed according to how well it predicts the response variable, and subsets that are either too big or ineffective in lowering the RMSE are penalized. The algorithm is pushed toward more condensed and significant feature subsets by this implicit penalization, which discourages high-dimensional solutions. Third, the meta-heuristic parallelizable structure of the algorithm maintains computing efficiency while enabling iterative convergence behavior to efficiently traverse large search spaces associated with high-dimensional data. This particular combination of optimizers with a regression-driven fitness strategy and parallelizable architecture has not, as far as we are aware, been integrated into any existing approach in the field of RF-sensed regression tasks. This positions HPA as a unique and scalable solution for high-dimensional regression problems.

Finally, research indicates that on RF-sensed datasets, HPA performs better than benchmark techniques. Specifically, HPA continuously yields higher $R^2$ values, lower RMSE, and lower MAE even as the feature space expands in size. These features demonstrate how HPA may effectively balance dimensionality reduction and model accuracy, making it dependable and scalable for high-dimensional data issues.

Figure 2 presents the basic flow and processing of the HPA. The variables X1 through X9, which are obtained from the different mechanisms of the GWO, ABC, and BA, indicate various positions calculated during the hybrid optimization process. In particular, the GWO mechanism is used to compute X1, X4, and X7, the ABC mechanism is used to compute X2, X5, and X8, and the BA method is used to compute X3, X6, and X9. A binary vector that represents a subset of features is represented by each of these variables; each bit in the binary vector indicates whether a feature is excluded (0) or selected (1) from the feature set. These variables are important because they can be iteratively optimized to find the most pertinent features for the regression problem.

Based on the outcomes of these mechanisms, the algorithm calculates three new positions (P1, P2, and P3) at each iteration. In order to keep the most pertinent features throughout the various optimization techniques, the final position is then calculated by taking the element-wise minimum of these three positions. By ensuring that only the best features are kept, this step reduces dimensionality and enhances model performance. Because they enable the feature subset to be refined over several iterations and let the HPA handle high-dimensional RF-sensed data, X1 through X9 are crucial to the feature selection process. The detailed explanation of process with mathematical representation of HPA's crucial stages is explained below.

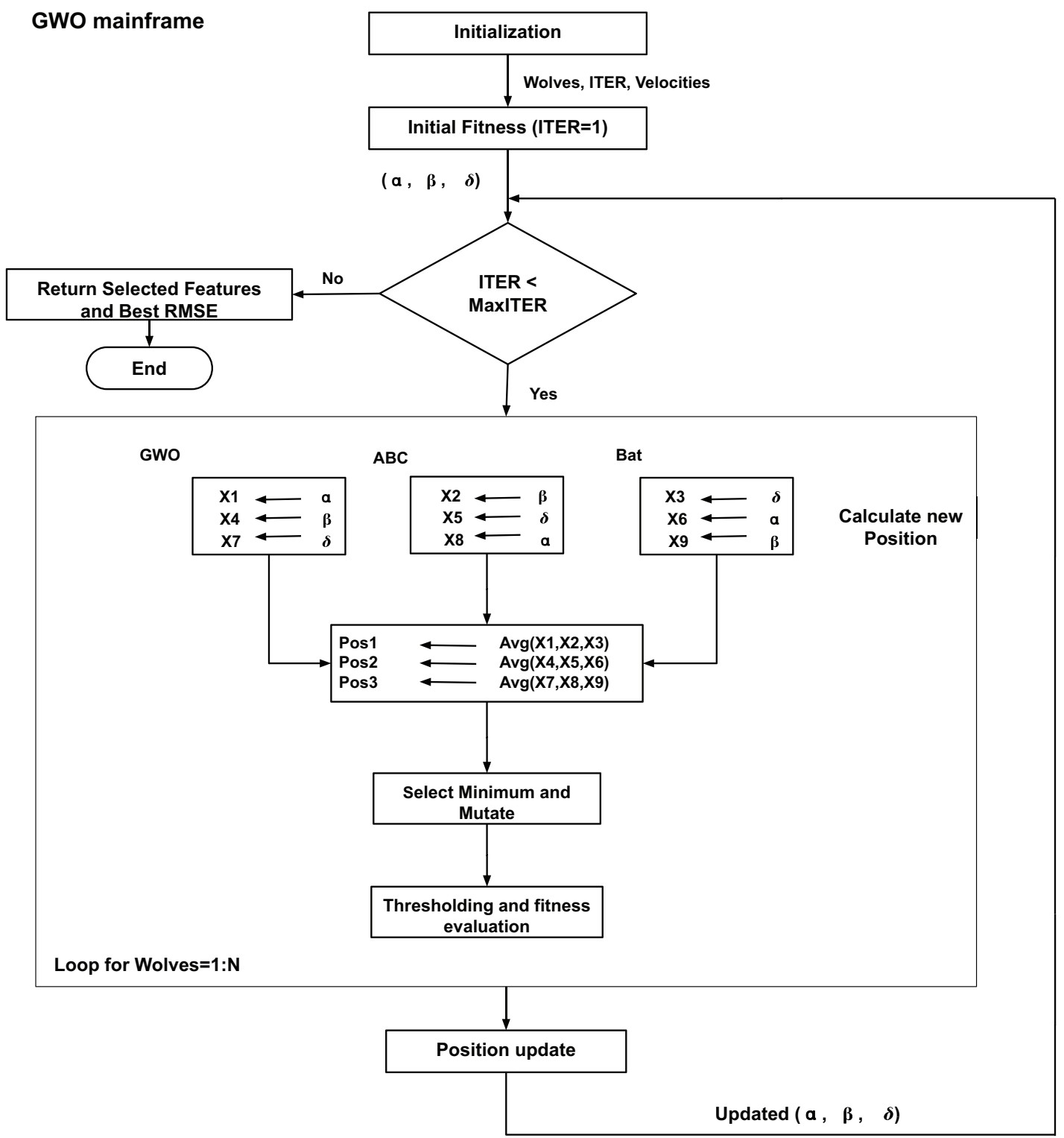

**Figure 2  Flowchart for the proposed HPA.**

The process begins by initializing a random population of wolves according to Eq. (1):

$$\text{wolves}_{i,j} \sim \mathscr{B}(0.5), \quad \text{for } i = 1, \dots, \text{num\_wolves}, \ j = 1, \dots, \text{num\_samples} \tag{1}$$

Each element $\text{wolves}_{i,j}$ in the initialization matrix is independently sampled from a Bernoulli distribution with a success probability of 0.5. The initial population goes through a fitness evaluation test. Equation (2) presents the fitness test in terms of RMSE:

$$\text{RMSE}(\text{wolves}_i) = \sqrt{\frac{1}{N} \sum_{k=1}^{N} (\hat{y}_k - y_k)^2}. \tag{2}$$

Following the fitness evaluation, the solutions are sorted based on RMSE $(\text{wolves}_i)$ in ascending order, where $\alpha_{\text{wolf}}$ is termed the best, $\beta_{\text{wolf}}$ the second best, and $\delta_{\text{wolf}}$ the third best. They are then subjected to an iterative position update, mutation and fitness evaluation process. The technique uses the main optimization loop and computes 9 new position variables X1 through X9. X1, X4, X7 use the main GWO position calculation mechanism while X2, X5 and X8 use the ABC mechanism and X3, X6 and X9 use the BA procedure respectively, as given by Eq. (3).

$$
\begin{aligned}
X_{1,4,7} &= \zeta_{\text{wolf},j} - A_1 \cdot \left| C_1 \cdot \zeta_{\text{wolf},j} - \text{wolves}_{i,j} \right|, \\
X_{2,5,8} &= \text{wolves}_{i,j} + \phi_1 \cdot \left( \text{wolves}_{i,j} - \zeta_{\text{wolf},j} \right), \\
X_{3,6,9} &= \text{wolves}_{i,j} + \left[ \phi_3 \cdot \left( \zeta_{\text{wolf},j} - \text{wolves}_{i,j} \right) + \text{velocities}_{i,j} \right].
\end{aligned}
\tag{3}
$$

Here, the term $\zeta_{\text{wolf},j}$ denotes a reference point derived from the leadership hierarchy within the population, selected based on the update index:

$$
\zeta_{\text{wolf},j} =
\begin{cases}
\alpha_{\text{wolf},j} & \text{if } X_{1,6,8}, \\
\beta_{\text{wolf},j} & \text{if } X_{2,4,9}, \\
\delta_{\text{wolf},j} & \text{if } X_{3,5,7},
\end{cases}
$$

- $X_{1,4,7}$: Positions updated *via* the GWO strategy.
- $X_{2,5,8}$: Positions adjusted using the ABC method.
- $X_{3,6,9}$: Positions refined through the BA formulation.
- $\alpha_{\text{wolf},j}, \beta_{\text{wolf},j}, \delta_{\text{wolf},j}$: Represent the top three agents (alpha, beta, delta) guiding the optimization in dimension $j$.
- $A_1, C_1$: Random modulation factors that influence exploration in GWO.
- $\phi_1$: Scaling coefficient applied in the ABC update mechanism.
- $\phi_3$: Scaling parameter used in the BA strategy.

For exploitation, three position variables, Pos1, Pos2 and Pos3 are computed by taking averages of different combinations of GWO, ABC and BA− as done in the GWO framework. This is given by Eq. (4):

$$
\begin{aligned}
P_1 &= \frac{X_1 + X_2 + X_3}{3}, \\
P_2 &= \frac{X_4 + X_5 + X_6}{3}, \\
P_3 &= \frac{X_7 + X_8 + X_9}{3}
\end{aligned}
\tag{4}
$$

**Algorithm 1  Proposed nature-inspired hybrid optimization algorithm (HPA).**

**Require:** objective_function: Evaluation function to be minimized
 1: num_samples: Dimensionality of the feature space
 2: lb: Lower bounds for each feature
 3: ub: Upper bounds for each feature
 4: num_wolves: Population size
 5: max_iterations: Maximum number of search cycles
**Ensure:** selected_samples: Best solution identified
 6: rmse: Fitness score of selected solution
 7: *wolves* ← InitializeWolves(*num_wolves*, *num_samples*)
 8: UpdateWolves(*wolves*, *objective_function*, *lb*, *ub*)                    ▷ Initial fitness evaluation
 9: SortWolves(*wolves*, *objective_function*)
10: **for** *iteration* = 1 **to** *max_iterations* **do**
11:     $a \leftarrow 2 - iteration \cdot (2/max\_iterations)$
12:     **for** $i = 1$ **to** *num_wolves* **do**
13:         UpdatePosition(*wolves[i]*, *alpha_wolf*, *beta_wolf*, *delta_wolf*, *a*, *lb*, *ub*)
14:     **end for**
15:     UpdateWolves(*wolves*, *objective_function*, *lb*, *ub*)
16:     SortWolves(*wolves*, *objective_function*)
17: **end for**
18: **Return** $\alpha\_wolf$ as selected_samples and its fitness as *rmse*

---

**Algorithm 2  UpdatePosition with velocity-driven movement.**

 1: **procedure** UPDATEPOSITION (*wolf*, *alpha_wolf*, *beta_wolf*, *delta_wolf*, *a*, *lb*, *ub*)
 2:     **for** $j = 1$ **to** *num_samples* **do**
 3:         % Grey Wolf dynamics
 4:         $X1 \leftarrow alpha\_wolf[j] - A1 \cdot |C1 \cdot alpha\_wolf[j] - wolf[j]| + velocities[i][j]$
 5:         $X4 \leftarrow beta\_wolf[j] - A1 \cdot |C1 \cdot beta\_wolf[j] - wolf[j]| + velocities[i][j]$
 6:         $X7 \leftarrow delta\_wolf[j] - A1 \cdot |C1 \cdot delta\_wolf[j] - wolf[j]| + velocities[i][j]$
 7:         % Artificial Bee Colony-like perturbation
 8:         $X2 \leftarrow wolf[j] + \phi \cdot (alpha\_wolf[j] - wolf[j]) + velocities[i][j]$
 9:         $X5 \leftarrow wolf[j] + \phi \cdot (beta\_wolf[j] - wolf[j]) + velocities[i][j]$
10:         $X8 \leftarrow wolf[j] + \phi \cdot (delta\_wolf[j] - wolf[j]) + velocities[i][j]$
11:         % Bat-inspired velocity and update
12:         $beta \leftarrow rand()$
13:         $velocities[i][j] \leftarrow velocities[i][j] + (delta\_wolf[j] - wolf[j]) \cdot beta$
14:         $max\_velocity \leftarrow 1$
15:         $velocities[i][j] \leftarrow \max(-max\_velocity, \min(velocities[i][j], max\_velocity))$
16:         $X3, X6, X9 \leftarrow wolf[j] + velocities[i][j]$

**Algorithm 2** (continued)

17:      % Averaged hybrid candidate positions
18:      $P1 \leftarrow (X1 + X2 + X3)/3$
19:      $P2 \leftarrow (X4 + X5 + X6)/3$
20:      $P3 \leftarrow (X7 + X8 + X9)/3$
21:      % Select the most promising estimate
22:      $wolf[j] \leftarrow \min(P1, P2, P3)$
23:      % Random mutation step
24:      **if** rand() $< \mu$ **then**
25:          $\varepsilon \leftarrow 0.01$
26:          $wolf[j] \leftarrow wolf[j] + \varepsilon \cdot (rand() - 0.5)$
27:      **end if**
28:      % Enforce boundary limits
29:      $wolf[j] \leftarrow \max(\min(wolf[j], ub[j]), lb[j])$
30:      % Thresholding to binary selection
31:      $wolf[j] \leftarrow 1$ if $wolf[j] > 0.5$, 0 otherwise
32:    **end for**
33:    $fitness \leftarrow objective\_function(wolf)$
34:    **return** $fitness$
35: **end procedure**

**Algorithm 3** **Fitness evaluation for entire population.**

1: **procedure** UPDATEWOLVES (*wolves, objective_function, lb, ub*)
2:    **for** $i = 1$ **to** *num_wolves* **do**
3:        $fitness\_values[i] \leftarrow$ UpdatePosition(*wolves[i], alpha_wolf, beta_wolf, delta_wolf, a, lb, ub*)
4:    **end for**
5: **end procedure**

Among these three position variables, the minimum is selected for each feature $j$ of each wolf $i$ using Eq. (5):

$$\text{wolves}_{i,j} = \min(P_1, P_2, P_3) \tag{5}$$

which then goes through mutation, Eq. (6). In this equation, the terms $\mu$ and $\varepsilon$ represents mutation probability and mutation strength respectively. The wolves positions are bounded by the boundary handling (bounded at lb = 0, ub = 1) Eq. (7), and binary quantization (thresholding at 0.5) is given by Eq. (8):

$$\text{wolves}_{i,j} = \text{wolves}_{i,j} + \varepsilon \cdot (\text{rand}() - 0.5) \quad \text{if rand}() < \mu \tag{6}$$

$$\text{wolves}_{i,j} = \max(\min(\text{wolves}_{i,j}, \text{ub}_j), \text{lb}_j) \tag{7}$$

$$\text{wolves}_{i,j} = \begin{cases} 1 & \text{if wolves}_{i,j} > 0.5 \\ 0 & \text{otherwise} \end{cases} \tag{8}$$

The Pseudo code for the technique is given by Algorithm 1, while the position update rules are given by Algorithms 2 and 3.

### Convergence analysis

It is generally accepted that performing convergence analysis of a NI process is arduous and often needless. There may be multiple reasons behind this conjecture: (1) They are often stochastic in nature, making their behaviour impossible to predict even with identical initial conditions (*Green, Aleti & Garcia, 2017*). (2) They usually involve complex interactions among multiple components, thereby creating nonlinear and unpredictable dynamics (*Shandilya, Datta & Nagar, 2023*). However, the convergence criteria may be defined, which in this case, is based on the following conditions:

1. Decreasing step size:

   $a$ must decrease linearly from 2 to 0 over iterations

2. Diversity maintenance:

   - Random components $\mathbf{r}_1$, $\mathbf{r}_2$, and $\phi$ ensure that the population maintains diversity
   - Mutation probability ensures that new potential solutions are explored

3. Boundary handling:

   $$\mathbf{X}_i = \max(\text{lb}, \min(\mathbf{X}_i, \text{ub})) \tag{9}$$

That is, the positions must be bounded within the lower and upper bounds [lb, ub]

4. Stability and convergence:

   - The algorithm should converge if the RMSE history stabilizes and does not show significant changes over iterations
   - Mathematically, the convergence condition is:

   $$\lim_{k \to \infty} f(\alpha^k) = f(\mathbf{X}^*) \tag{10}$$

where $k$ is the iteration number, $\alpha^k$ is the position of the alpha wolf at iteration $k$, and $\mathbf{X}^*$ is the global optimum solution minimizing the RMSE

The convergence behavior of the proposed HPA algorithm, as illustrated in Fig. 3. The first plot shows that HPA consistently achieves lower RMSE values across iterations, indicating a faster and more stable convergence toward the optimal solution. The second plot, which depicts the rate of change in RMSE, highlights HPA's smooth and gradual performance improvements with minimal fluctuations. which further emphasizes its convergence stability. To validate these observations, we conducted multiple independent runs, all of which confirmed HPA's robust and consistent behavior across different initializations. These findings affirm the reliability of HPA in high-dimensional
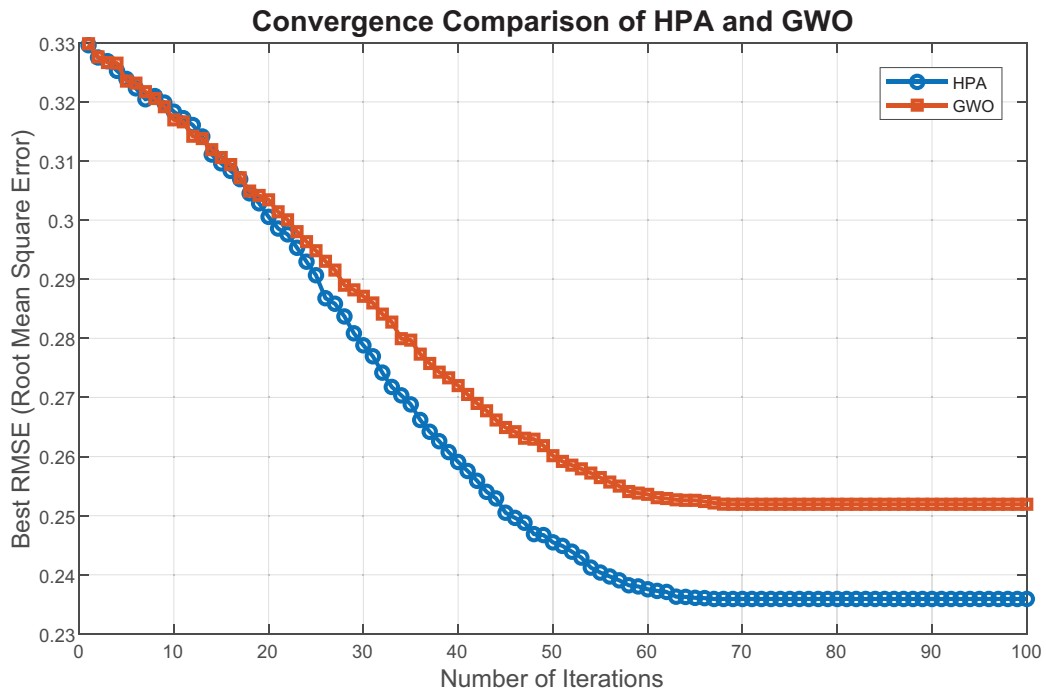

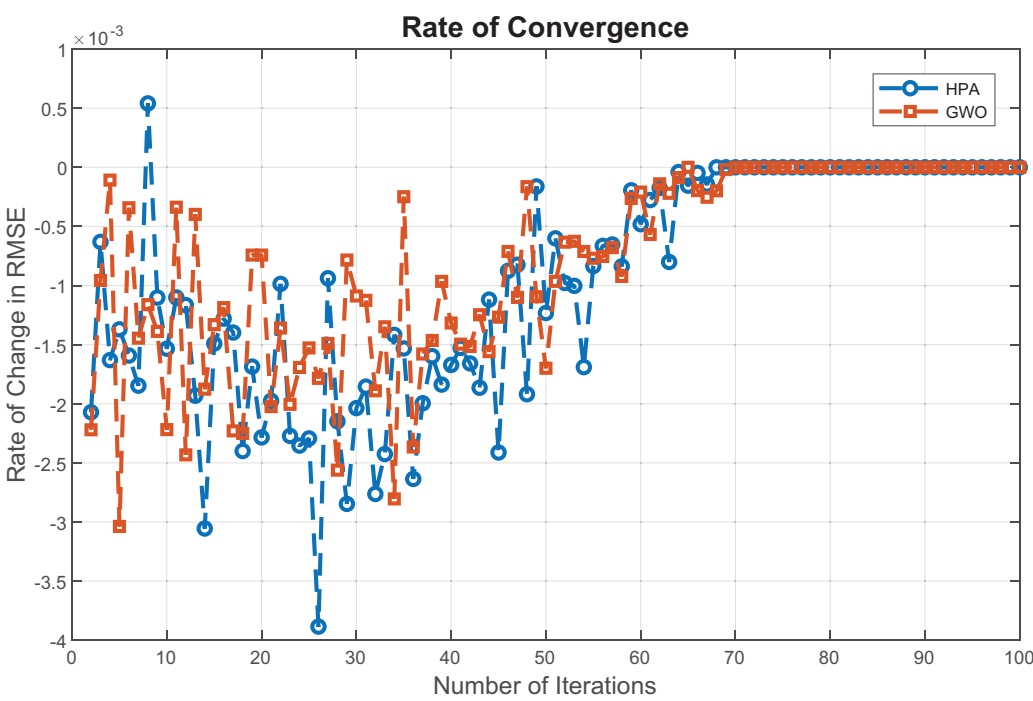

**Figure 3 Convergence analysis of HPA and GWO.**

**Table 3 Comparison of time complexities of HPA and baseline algorithms.**

| Algorithm | Per-iteration complexity | Overall complexity |
|---|---|---|
| GWO | $O(\alpha \cdot (\beta + f(\beta)) + \alpha \log \alpha)$ | $O(\texttt{max\_iterations} \cdot \alpha \cdot (\beta + f(\beta) + \log \alpha))$ |
| ABC | $O(\alpha \cdot \beta + \alpha \cdot f(\beta))$ | $O(\texttt{max\_iterations} \cdot \alpha \cdot (\beta + f(\beta)))$ |
| BA | $O(\alpha \cdot \beta + \alpha \cdot f(\beta))$ | $O(\texttt{max\_iterations} \cdot \alpha \cdot (\beta + f(\beta)))$ |
| **HPA** | $O(\alpha \cdot (8 \cdot \beta + f(\beta)) + \alpha \log \alpha)$ | $O(\texttt{max\_iterations} \cdot \alpha \cdot (8 \cdot \beta + f(\beta) + \log \alpha))$ |

optimization settings and underscore its effectiveness for regression-based feature selection, where stable convergence is essential for performance and reproducibility.

### Complexity analysis

This section provides a detailed analysis of the time complexity of the proposed HPA and its comparison with baseline algorithms, including GWO, ABC, and BA.

The GWO serves as the foundation for the HPA, which incorporates improvements from the ABC and BA. Although new position update calculations are included, they are carried out using the same iteration framework and are linear in relation to the number of samples and wolves.

**Initialization phase:**

$$O(\alpha \times \beta) + O(\alpha \times f(\beta)),$$

where $\alpha$ is the population size, $\beta$ is the number of features, and $f(\beta)$ is the complexity of fitness evaluation.

**Iterative optimization phase:**

The primary components of the iterative phase are:

- Position updates using GWO as the foundational framework: $O(\alpha \times \beta)$,
- Enhancements from ABC and BA, implemented as linear operations: $O(2 \times \alpha \times \beta)$,
- Combination, mutation, and binary discretization: $O(5 \times \alpha \times \beta)$,
- Fitness evaluation and leader updates: $O(\alpha \times f(\beta)) + O(\alpha \log \alpha)$.

The total complexity per iteration is:

$$O(\alpha \times (8 \times \beta + f(\beta)) + \alpha \log \alpha).$$

Several linear operations carried out during position updates, combinations, and improvements are combined to provide the constant factor "8" in this case. The "8" offers a thorough analysis of these contributions, but it's crucial to remember that constant components have no bearing on the asymptotic growth rate. Therefore, in theoretical terms, the complexity remains $O(\alpha \times (\beta + f(\beta)) + \alpha \log \alpha)$, comparable to GWO.

For `max_iterations` iterations, the overall complexity is:

$$O(\texttt{max\_iterations} \times \alpha \times (8 \times \beta + f(\beta) + \log \alpha)).$$

Table 3 provides a comparative analysis of the time complexities of HPA and the baseline algorithms.

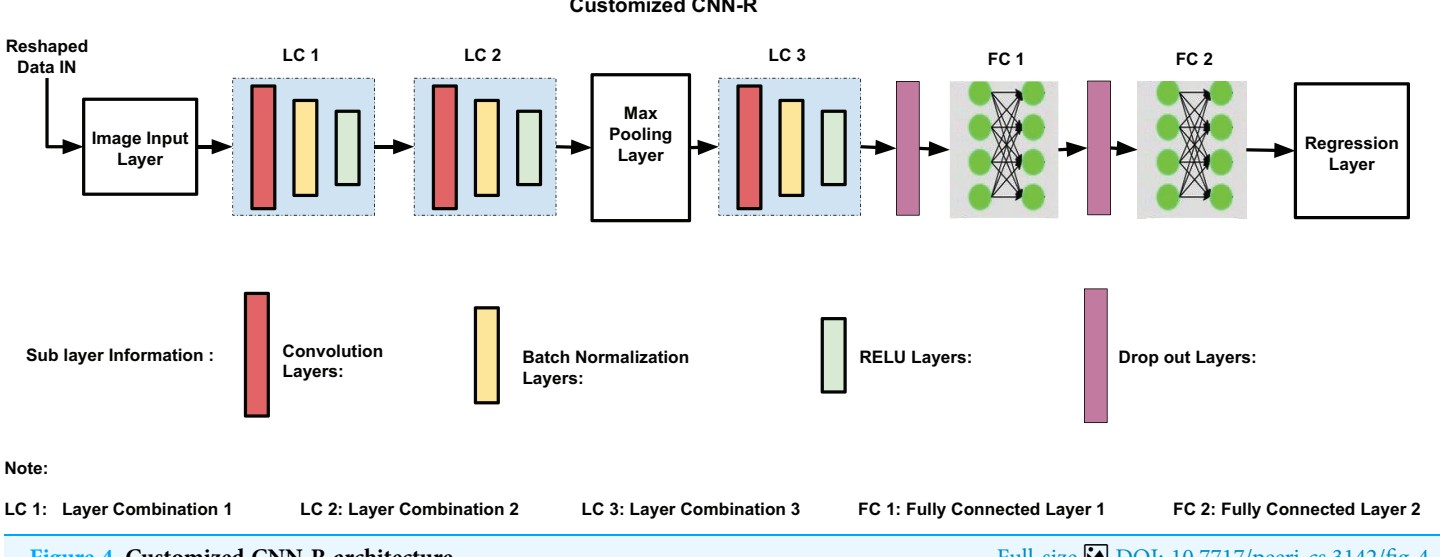

**Figure 4** Customized CNN-R architecture.   

According to Table 3, the temporal complexity of HPA remains comparable to GWO. The computational overhead introduced by the ABC and BAT components is negligible. These enhancements do not require new loops or higher-order dependencies, thus preserving the algorithm's computational efficiency.

## Customized CNN-R model architecture

The backbone of CNN-R is identical to the one used in *Ali et al. (2025)*, except for the final layers tailored for regression tasks. With 17 layers and 16 connections, the customized CNN-R model architecture perfectly balances computational economy and performance. As shown in Fig. 4, the architecture consists of three primary layer combinations (LC1, LC2, and LC3). Convolutional, batch normalization, and ReLU layers are among these combinations; pooling, dropout, and fully connected layers come next. Every component is designed to improve the model's training effectiveness, generalization ability, and accuracy.

Every layer in the CNN-R architecture is essential to enhancing the performance of the model. In order to adapt dynamically to different datasets and reduce computing costs, the input layer is designed to support feature combinations produced during the feature selection process. Different levels of characteristics are captured by the layer combinations LC1, LC2, and LC3, which use convolutional layers with filter widths of 64, 128, and 256, respectively. In order to recognize complex patterns, ReLU activation functions incorporate non-linearity, whereas batch normalization standardizes inputs to stabilize and speed up training. Following LC2, dropout layers with a 0.5 probability improve generalization by avoiding overfitting, while max pooling minimizes computational expenses and spatial dimensions while preserving crucial information.

To determine how max pooling and average pooling affected regression performance indicators, particularly RMSE, a study of the CNN-R model's pooling layers was carried

out. Two CNN-R architectures were created, one with a max pooling layer and the other with an average pooling layer, in order to determine the effect of layer variation. Superior RMSE values (0.083 *vs.* 0.095) demonstrated that max pooling, which chooses the maximum value from feature regions, performed noticeably better than average pooling. The ability of max pooling to highlight important features which are essential for high precision tasks is credited with this superiority. On the other hand, average pooling resulted in somewhat low performance even if it provided smoother feature representations. The CNN-R model's preference for max pooling is confirmed by the RMSE results, which indicate that it is the best option for applications requiring accurate feature extraction.

Compared to deeper networks like GoogLeNet or ResNet-50, the CNN-R model's small and effective architecture provides a number of design advantages. Combinations of layers that are stacked preserve accuracy while using less computing power and training time. Using $1 \times 1$ kernels allows for extensive feature extraction with a modest processing overhead. The model's ability to identify different patterns is improved by the hierarchical filter organization in LC1, LC2, and LC3, and its generalization ability is further increased by the addition of dropout and batch normalization layers.

Despite being simpler, the CNN-R model exhibits competitive accuracy and training speed when compared to typical CNN architectures. For real-time applications or resource-constrained contexts where computational efficiency is essential, its tailored architecture is perfect. Combining the CNN-R model with the HPA yields estimation accuracy comparable to the GRU, as shown in 'Results', which makes it appropriate for numeric datasets.

## Combined framework

The proposed framework is shown in Fig. 5. It consists of three major parts: The first part describes the machine learning implementation without the use of any feature selection technique. In initialization stage, the dataset is accessed. For supervised learning, the response value is identified by computing the dielectric properties derived from S-parameters, following the method described by *Altherwy & McCann (2020)*. In data pre-processing stage, the data is normalized and training (TD) and testing (TSD) data are identified. Three deep learning models, LSTM, Bi-LSTM and GRU are trained and tested with training and testing data, and predictions are made. During this process, the hyperparameters for these three models are kept same for consistency and fairness. Predictions are made by these models, and performance is evaluated in-terms of the selected performance metrics. The best performing model is selected for further analysis in the second part.

In second part, labeled as Feature Selection with NI-FS technique in Fig. 5, the NI feature selection techniques are incorporated after the pre-processing stage, and best features are selected against each technique. Subsequently, a sub dataset is created comprising the best features and corresponding responses. The best model from part 1 is used with the same hyperparameter configurations. In this step, seven benchmark NI feature selection techniques are used, the model is trained and tested with sub dataset

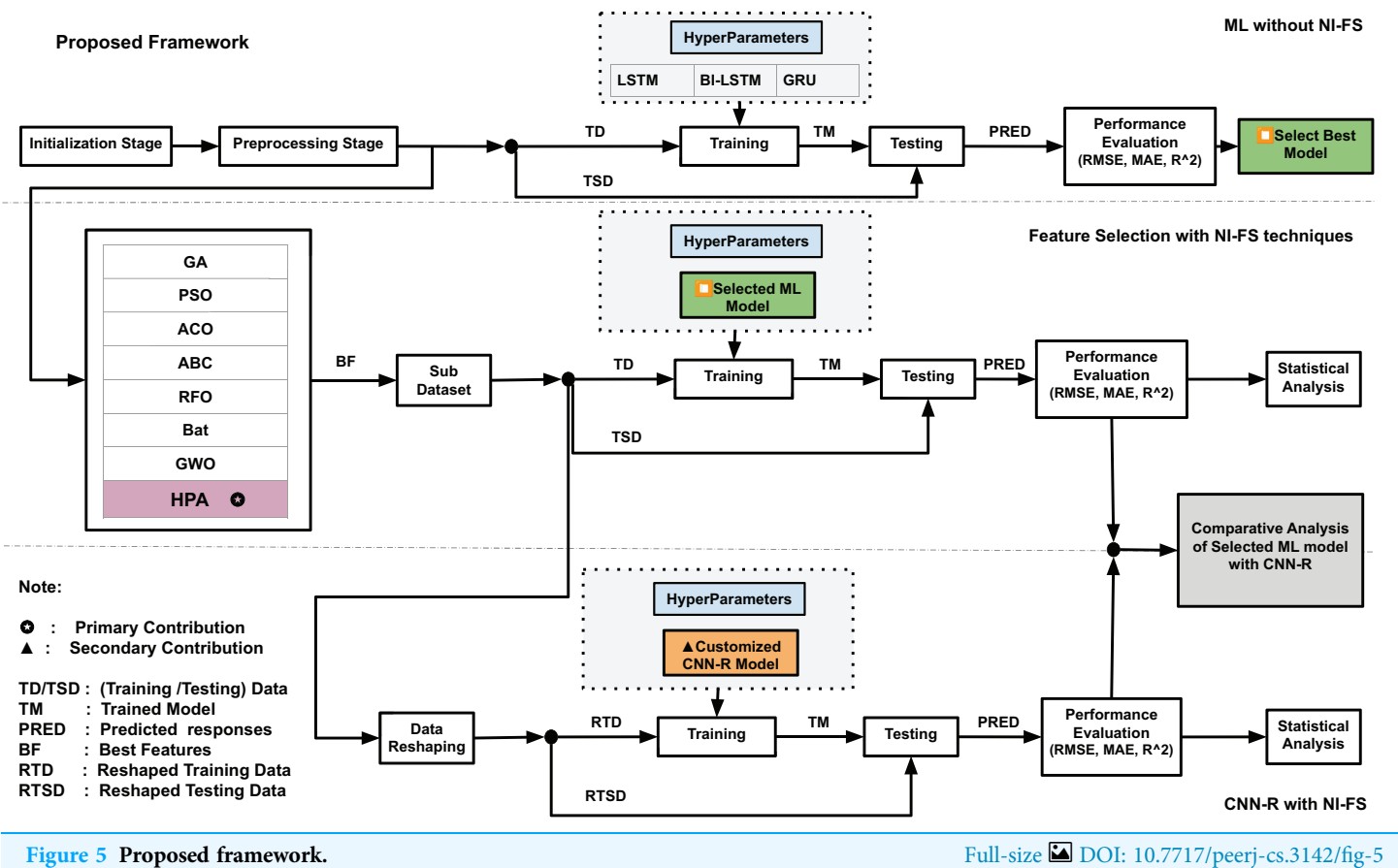

**Figure 5 Proposed framework.**

obtained with each technique, and the performance is evaluated. The HPA is then integrated within this step and the process is repeated to get the performance metric. The statistical analysis is subsequently performed.

In third part, the customized CNN model, described in Fig. 4, is deployed after reshaping the data obtained from the sub-dataset. The hyperparameters for the CNN-R model are identified, the model is trained and tested to get the performances parameters, and statistical analysis is performed. The comparative analysis among the results of part 2 and part 3 are also performed to see the impacts on performance gains in terms of RMSE, MAE and $R^2$. The process gives us insights of any impacts of switching from GRU (the best performing model) to CNN-R.

The hyperparameter settings for the benchmark feature selection techniques along with HPA are shown in Table 4.

The hyperparameter settings for ML model simulations are given in Table 5.

The number of neurons is kept at 2,000. The execution environment is set to 'GPU'. GTH stands for gradient threshold, where ILR, LRDP and LRDF are initial learning rate, learning rate drop period, and learning rate drop factor, respectively. OPT stands for optimizer which is set to ADAM. The layered structure of the LSTM, Bi-LSTM and GRU are kept same as used in the baseline work (*Altherwy et al., 2024*).

**Table 4 Hyperparameter settings for benchmark feature selection techniques.**

| Algorithm | Population size | Max iterations | Other key hyperparameters |
|---|---|---|---|
| ABC | 20 employed bees | 100 | — |
| ACO | 20 ants | 100 | $\alpha = 1$, $\beta = 2$, pheromone evaporation = 0.5, initial pheromone = 0.1 |
| BAT | 20 bats | 100 | $A_0 = 0.5$, decay $\alpha = 0.5$, emission rate $\gamma = 0.5$ |
| GA | 20 individuals | 100 generations | Crossover = 0.8, mutation = 0.01 |
| PSO | 20 particles | 100 | — |
| RFO | 20 foxes | 100 | — |
| GWO | 20 wolves | 100 | — |
| **HPA** | 20 agents | 100 | Hybrid of GWO, ABC, and BAT strategies |

**Table 5 Hyperparameters for the proposed LSTM/GRU and CNN-R framework.**

| Framework | Epoch | GTH | ILR | LRDP | LRDF | Optimizer |
|---|---|---|---|---|---|---|
| LSTM/Bi-LSTM/GRU | 100 | 1 | Piecewise | 50 | 0.5 | Adam |
| CNN-R | 100 | N/A | 0.001 | N/A | N/A | Adam |

## RESULTS

In this section, we discuss the results of the feature selection framework for grape moisture estimation. The framework is implemented in MATLAB 2022A (The MathWorks, Natick, MA, USA), on a system with INTEL CORE i7 processor and 32 GB RAM, equipped with NVIDIA TESLA K40 GPU. To evaluate the framework we compute performance metrics, namely RMSE, MAE and $R^2$ values. The optimizers are used for feature selection and then we use two ML models for moisture estimation.

### Feature subset size and efficiency evaluation

In this section, we evaluate the feature subset size and the run-time efficiency of the benchmark feature selection techniques, including ABC, BAT, GWO, and HPA. Table 6, tells the details about selected feature subset by HPA along with individual benchmark techniques contributing in the design of HPA. Table 7, presents the elapsed time for processing feature selection technique in main pipeline.

The percentage reduction in features is calculated based on the initial number of features (48,030), indicating the extent of dimensionality reduction achieved by each technique. As seen, GWO and HPA result in significant reductions, selecting only a small fraction of the original feature set.

Table 7, provides insights into the computational time required for each feature selection technique. While ABC and BAT are relatively efficient, with run-times of approximately 140 s, HPA requires 161 s comparable to GWO. The GWO took 149 s for execution.

These run-time values correlate with the computational complexity of the algorithms, as higher-dimensional search spaces typically require more time for exploration. The HPA algorithm, being a hybrid, involves more complex operations due to its integration of multiple techniques, explaining its higher run-time but still comparable with GWO.

Table 6 Selected feature subset size and percentage reduction.

| Feature selection technique | Number of selected features | Percentage reduction (%) |
| --- | --- | --- |
| ABC | 24,045 | 49.99 |
| BAT | 24,064 | 49.98 |
| GWO | 3,381 | 93.02 |
| HPA | 2,417 | 95.03 |

Table 7 Run-time efficiency of feature selection techniques.

| Feature selection technique | Run-time (seconds) |
| --- | --- |
| ABC | 141.8581 |
| BAT | 139.3210 |
| GWO | 149.3698 |
| HPA | 161.3247 |

In Table 6, we see that GWO and HPA result in a substantial reduction in feature size, achieving 93% and 95% feature reduction, respectively. While ABC and BAT also reduce the feature set by about 50%, they retain more features compared to GWO and HPA.

In Table 7, the run-time performance suggests that while ABC and BAT are relatively fast, HPA, which selects fewer features, has the comparable computational cost. This could be attributed to the hybrid nature of HPA, which combines the strengths of multiple algorithms, leading to higher computational cost. GWO, while reducing features significantly, also takes considerable time, although less than HPA.

## Estimation using GRU and benchmark techniques

Table 8 presents the performance metrics of the GRU model using various benchmark NI feature selection techniques, including GA, ACO, PSO, ABC, RFO, BA, and GWO. These results sets a benchmark for quantify our HPA's performance.

For RMSE, the GWO technique achieves the lowest value (0.256), indicating the highest accuracy, followed by GA (0.302) and PSO (0.308). In terms of MAE, GWO again outperforms the other techniques with the lowest error (0.193), followed by GA (0.226) and ACO (0.234). The $R^2$ values indicate that GWO provides the best fit to the data (0.928), surpassing all other techniques, with ABC (0.908) and GA (0.907) trailing behind.

These findings show that GWO, the benchmark method, is very successful in choosing the best features for the GRU model, producing a high $R^2$, low RMSE, and MAE. In the next sections, we will demonstrate that HPA provides an even more efficient feature selection procedure by hybridizing GWO, ABC, and BAT. By balancing exploration and exploitation, HPA improves model performance by including more nature-inspired methods, which also increases accuracy and lowers error levels.

## Estimation using CNN-R and benchmark techniques

Table 9 presents the performance metrics of the CNN-R model using the benchmark NI feature selection techniques.

**Table 8  Performance metrics of GRU model using benchmark techniques.**

| Model | Metric | GA | ACO | PSO | ABC | RFO | BA | GWO |
|-------|--------|-----|-----|-----|-----|-----|-----|-----|
| GRU | RMSE | 0.302 | 0.312 | 0.308 | 0.309 | 0.347 | 0.312 | 0.256 |
| | MAE | 0.226 | 0.234 | 0.235 | 0.235 | 0.267 | 0.237 | 0.193 |
| | $R^2$ | 0.907 | 0.905 | 0.906 | 0.908 | 0.904 | 0.903 | 0.928 |

**Table 9  Performance metrics of CNN-R model using benchmark techniques.**

| Model | Metric | GA | ACO | PSO | ABC | RFO | BA | GWO |
|-------|--------|-----|-----|-----|-----|-----|-----|-----|
| CNN-R | RMSE | 0.034 | 0.058 | 0.085 | 0.048 | 0.090 | 0.044 | 0.084 |
| | MAE | 0.057 | 0.068 | 0.096 | 0.037 | 0.090 | 0.078 | 0.094 |
| | $R^2$ | 0.998 | 0.999 | 0.998 | 0.998 | 0.997 | 0.998 | 0.997 |

For RMSE, the GA technique achieves the lowest value (0.034), indicating the highest accuracy, followed closely by BA (0.044) and ABC (0.048). In terms of MAE, ABC outperforms the other techniques with the lowest error (0.037), followed by GA (0.057) and ACO (0.068). The $R^2$ values indicate that ACO provides the best fit to the data (0.999), slightly surpassing other techniques, with GA, PSO, ABC, and BA all achieving $R^2$ values of 0.998.

Overall, the GA and ABC techniques demonstrate superior performance across the RMSE and MAE metrics, making them the most effective NI feature selection techniques for optimizing the CNN-R model in this study. The consistently high $R^2$ values across all techniques confirm that the CNN-R model provides an excellent fit to the data, regardless of the specific feature selection technique used.

## Performance analysis of the proposed technique (HPA)

This section's empirical data provides a graphic representation of how HPA outperforms the individual baseline approaches in terms of accuracy and performance, obtaining superior performance metrics in both the CNN-R and GRU models. Figure 6 represents the RMSE values of the ML models when integrated with benchmark and HPA. Similarly, Fig. 7 represent the MAE statistics and Fig. 8 represents the $R^2$ values of the ML models when integrated with the benchmark techniques and HPA. The GRU model's performance when paired with the benchmark methods and then with the HPA shows notable variances in terms of RMSE. With an incredibly low RMSE of 0.02, the HPA stands out and contributes a notable improvement in model accuracy. Compared to the best among the benchmark techniques, the HPA promises around 92% improvement in the estimation accuracy. This investigation demonstrates the potential of hybrid optimizers in delivering unparalleled estimation performance in the GRU models.

When taking MAE into consideration for the GRU model, the HPA leads to an exceptionally low MAE of 0.013, indicating a notable improvement in estimation accuracy. With an MAE of 0.193, GWO was the best-performing traditional optimizer; by contrast, the HPA ensures an impressive 93% improvement. Similarly, the HPA achieves almost the perfect result of 0.999 in terms of $R^2$, which is around 7.7% improved compared to the best benchmark method, GWO.
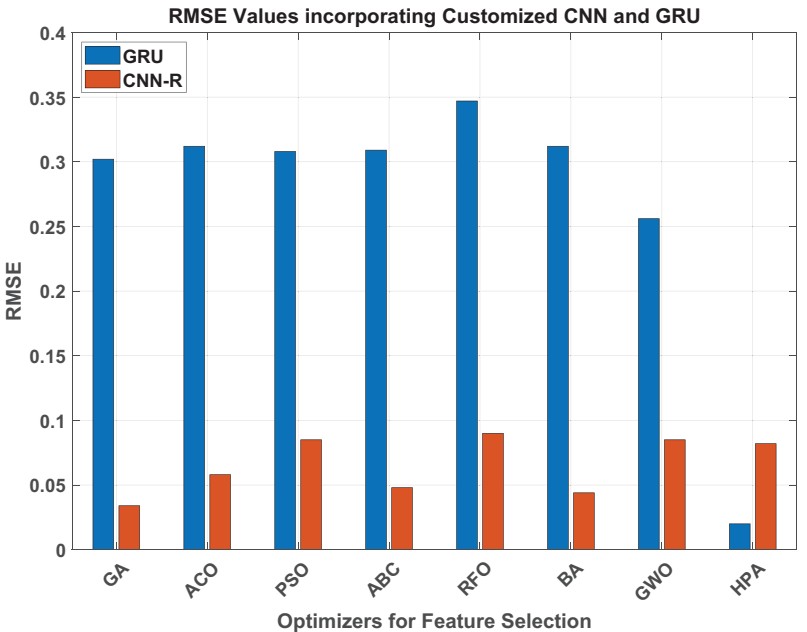

**Figure 6  RMSE values.**

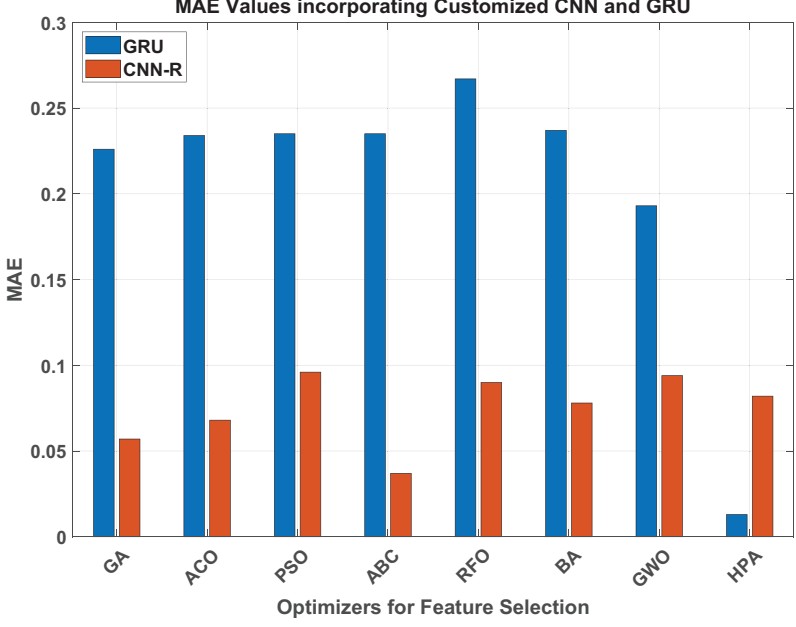

**Figure 7  MAE values.**

## Performance analysis of the CNN-R

Considering RMSE values across various optimizers, a significant performance improvement is evident when transitioning from the GRU model to the customized CNN-R. For instance, the GA optimizer shows a remarkable reduction in RMSE from 0.302 with GRU to 0.034 with CNN-R, demonstrating an improvement of approximately

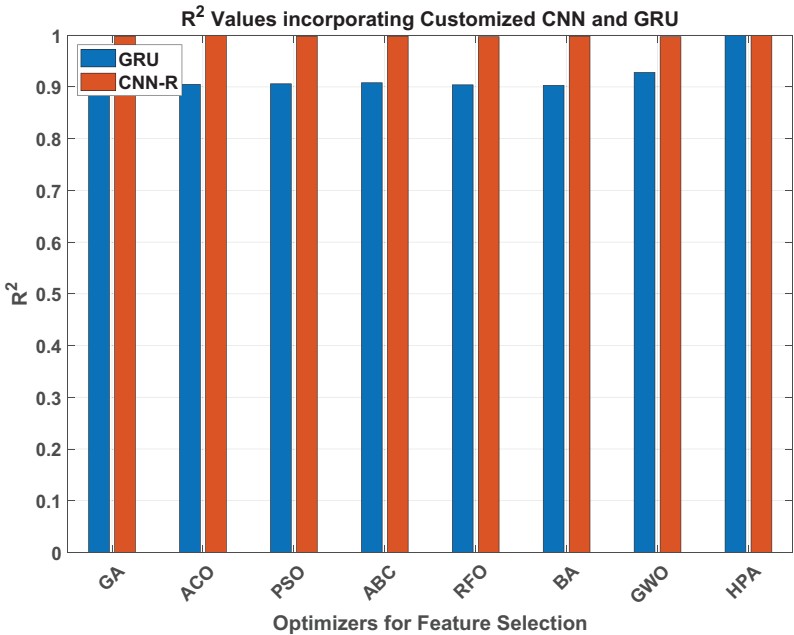

**Figure 8** $R^2$ **values.**               

88.7%. Similarly, the ACO optimizer reduces RMSE from 0.312 with GRU to 0.058 with CNN-R, achieving an improvement of around 81.4%. PSO and ABC optimizers also exhibit substantial gains, with RMSE reductions from 0.308 to 0.085 and from 0.309 to 0.048, corresponding to improvements of 72.4% and 84.5%, respectively. Even the least effective optimizers for GRU, such as RFO and BA, show significant enhancements with CNN-R, with RMSE decreasing from 0.347 to 0.09 and from 0.312 to 0.044, reflecting improvements of 74% and 85.9%, respectively. The GWO optimizer with GRU manages an RMSE of 0.256 and with CNN-R of 0.084, which shows an improvement by about 67.2%.

The HPA, which already excels with the GRU model at an RMSE of 0.02, maintains its high performance with the CNN-R model, achieving an RMSE of 0.082. Although, during this transition, the proposed method does not bring about any improvement in terms of RMSE, due to its already optimized performance, the consistently low RMSE values highlight its robustness across different models. This analysis underscores the transformative impact of adopting the CNN-R architecture, which consistently outperforms the GRU model across all benchmark optimizers, achieving substantial reductions in RMSE. It highlights the CNN-R model's superior accuracy and predictive performance in machine learning applications, with the HPA continuing to demonstrate its effectiveness as a top-performing optimizer.

Similarly, considering MAE and $R^2$ values for both GRU and CNN-R models across various optimizers, a significant performance improvement is evident when transitioning from the GRU model to the customized CNN-R. For instance, the GA optimizer shows a notable reduction in MAE from 0.226 with GRU to 0.057 with CNN-R, demonstrating an improvement of approximately 74.8%. Similarly, all the other optimizers reduce MAE and

increase the $R^2$ values by a subtantial margin. The HPA, however, once again maintains reasonable numbers against both MAE and $R^2$ parameters.

Aditionally, A comparison of the $R^2$ values on the training and testing data was also carried out in order to guarantee the robustness of system and address concerns about possible overfitting. With a very minimal difference between training and testing $R^2$, the model obtained nearly flawless values for both. Because there is no considerable decrease in performance when applied to unseen data, this suggests that the model generalizes effectively and is not overfitting. In order to confirm that the high performance is consistent and not the consequence of overfitting to a particular subset of the data, 10-fold cross-validation was also used to further evaluate the model's stability across several data splits.

In conclusion, the comparison of the GRU and CNN-R models across various optimization techniques reveals distinct advantages and performance characteristics. The GRU model, while effective, shows notable improvements in accuracy metrics (RMSE, MAE, $R^2$) when replaced by the CNN-R architecture. Across optimizers such as GA, ACO, PSO, ABC, RFO, BA, GWO, and the HPA, the CNN-R consistently achieves lower error metrics (RMSE, MAE) and higher accuracy ($R^2$), showcasing its superior predictive capability.

Specifically, the CNN-R model demonstrates significant reductions in error metrics compared to GRU, with improvements ranging from approximately 59% to 88.7% in RMSE and from 51.3% to 84.8% in MAE across various optimizers. The $R^2$ values also exhibit improvements ranging from 7.8% to 10.9% with CNN-R consistently achieving higher values, indicating better fit and predictive power.

Furthermore, the proposed HPA for feature selection stands out for its exceptional performance across both GRU and CNN-R models, maintaining consistently low error metrics and high $R^2$ values. This underscores the effectiveness of hybrid optimization techniques in enhancing model performance.

## Analysis with previous work, LSTM and Bi-LSTM networks

The effectiveness of the HPA can be observed by having comparative analysis of our findings with the previous work and also with models like LSTM and Bi-LSTM. Our findings are highlighted in Table 10. In our previous work, GRU coupled with PSO provided best results in terms of RMSE, MAE and $R^2$ values. While the values are self-explanatory, two important conclusions that may be drawn from the table are as follows:

1. When coupled with the proposed feature selection method, each of the three models, LSTM, Bi-LSTM and GRU exhibit improved values of RMSE, MAE and $R^2$.
2. Confirming our earlier claim, GRU, with and without the feature selection method stands out among the three models.

## Statistical analysis

This section presents the detailed statistical analysis by presenting ANOVA statistics in Table 11 and box plot analysis of RMSE, MAE and $R^2$ values in Figs. 9, 10 and 11, respectively.

**Table 10 Performance comparison of previous work (*Altherwy et al., 2024*) and with HPA, along with percentage impact.**

| Model | Previous work | | | With HPA | | | Percentage impact | | |
|---|---|---|---|---|---|---|---|---|---|
| | RMSE | MAE | $R^2$ | RMSE | MAE | $R^2$ | RMSE | MAE | $R^2$ |
| LSTM | 0.301 | 0.230 | 0.900 | 0.053 | 0.0403 | 0.997 | 82.39 | 81.30 | 9.7 |
| Bi-LSTM | 0.297 | 0.228 | 0.900 | 0.042 | 0.093 | 0.998 | 85.85 | 59.21 | 9.8 |
| GRU | 0.290 | 0.220 | 0.920 | 0.020 | 0.013 | 0.999 | 93.10 | 94.09 | 7.9 |

**Table 11 ANOVA statistic table.**

| Parameter | Source | SS | df | MS | F | Prob > F |
|---|---|---|---|---|---|---|
| RMSE | Columns | 0.1681 | 1 | 0.1681 | 29.57 | $8.74 \times 10^{-5}$ |
| | Error | 0.0796 | 14 | 0.0057 | | |
| | Total | 0.2477 | 15 | | | |
| MAE | Columns | 0.0673 | 1 | 0.0673 | 19.70 | $6.00 \times 10^{-4}$ |
| | Error | 0.0479 | 14 | 0.0034 | | |
| | Total | 0.1152 | 15 | | | |
| $R^2$ | Columns | 0.0243 | 1 | 0.0243 | 44.90 | $1.01 \times 10^{-5}$ |
| | Error | 0.0076 | 14 | 0.0005 | | |
| | Total | 0.0319 | 15 | | | |

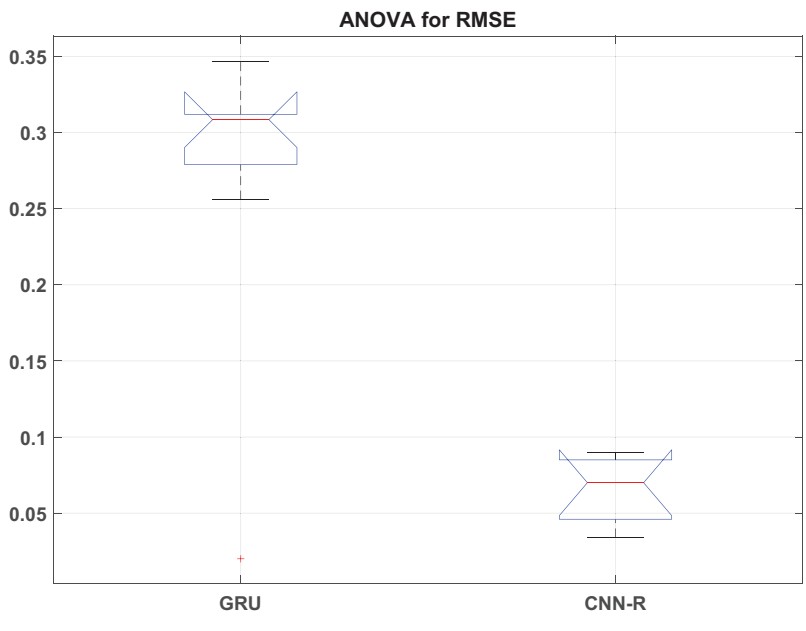

**Figure 9 RMSE box plot.**

The ANOVA results reveal significant differences in performance metrics (RMSE, MAE, and $R^2$) between the GRU and CNN-R models across various optimizers. For RMSE, the F-statistic of 29.57 with a *p*-value of $8.74 \times 10^{-5}$ indicates strong evidence that the

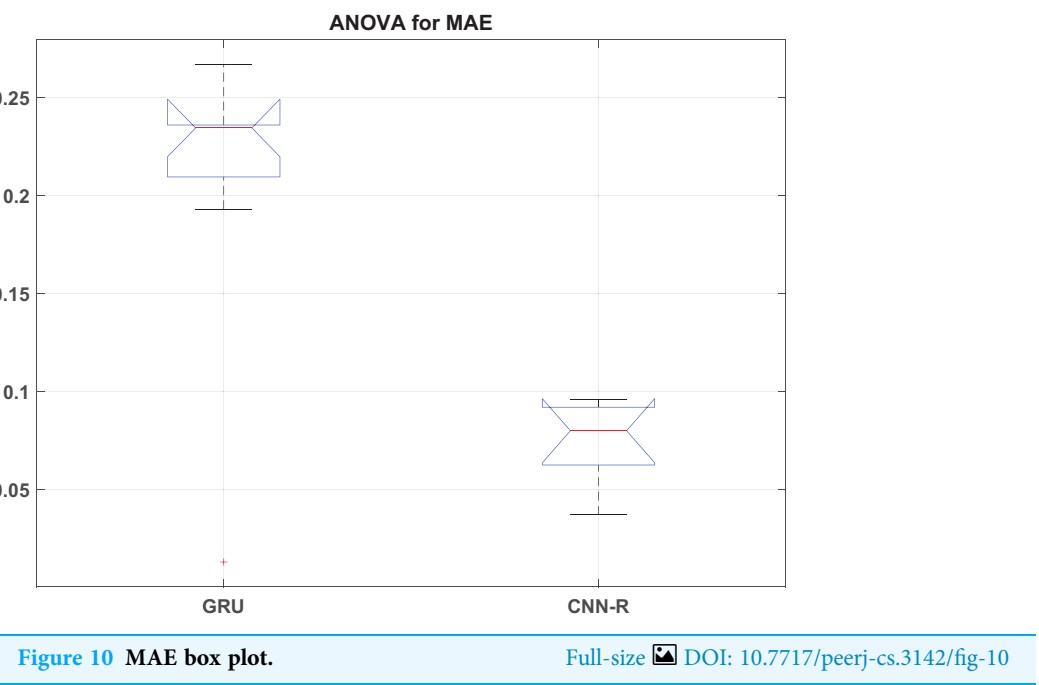

**Figure 10  MAE box plot.**     

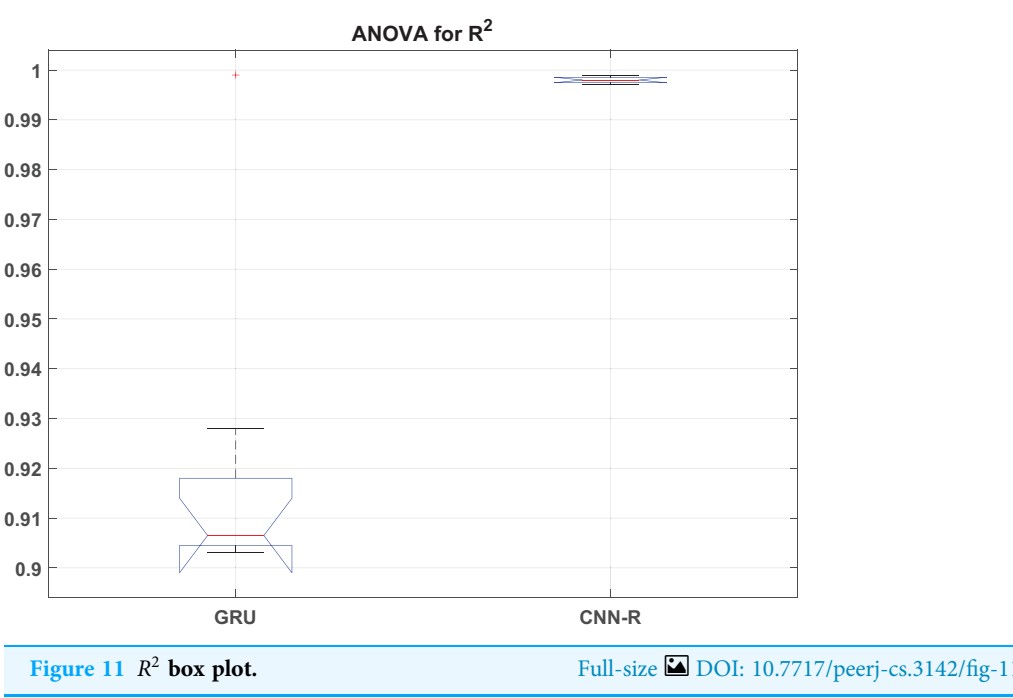

**Figure 11  $R^2$ box plot.**     

RMSE values vary significantly across different models and their associated optimizers. Similarly, MAE shows an F-value of 19.70 with a *p*-value of $6.00 \times 10^{-4}$, affirming substantial differences in MAE between models. Additionally, $R^2$ exhibits the highest variability among the metrics, with an F-value of 44.90 and a very low *p*-value of $1.01 \times 10^{-5}$, underscoring robust differences in coefficient of determination across the models.

These findings underscore the superior predictive performance of the CNN-R model compared to GRU across all tested metrics. The CNN-R consistently achieves lower RMSE and MAE values, indicating more accurate predictions and reduced errors compared to the GRU model. Moreover, $R^2$ values for the CNN-R model are significantly higher, suggesting better model fit and a stronger relationship between predicted and actual outcomes. The statistical significance of these results, as indicated by the low $p$-values, reinforces the conclusion that the CNN-R architecture, along with its optimized feature selection techniques, is better suited for tasks requiring precise and reliable predictions.

In practical terms, these insights support the preference for adopting CNN-R over GRU in machine learning applications where accuracy and reliability are paramount. The demonstrated improvements in predictive metrics validate the effectiveness of advanced neural network architectures and hybrid optimization techniques, such as the one proposed in this work, in enhancing model performance. This analysis provides a clear statistical basis for decision-making in model selection and optimization, ensuring optimal outcomes in real-world applications of machine learning and predictive analytics.

## CONCLUSION

This research explored the performance of GRU and customized CNN-R model across various nature-inspired optimizers for feature selection, including a novel hybrid optimizer. Our simulation results, supported by statistical analysis, demonstrated significant differences in prediction accuracy and model fit between the GRU and CNN-R architectures. The CNN-R model consistently outperformed the GRU model in terms of RMSE, MAE, and $R^2$, achieving lower error metrics and higher coefficients of determination across all tested optimizers.

The ANOVA results reinforced these findings, showing statistically significant differences with $p$-values well below the 0.05 threshold for RMSE (0.0000874), MAE (0.0006), and $R^2$ (0.0000100926). These low $p$-values indicate that the observed performance improvements are highly unlikely to be due to chance. Specifically, the CNN-R model demonstrated substantial reductions in RMSE and MAE, and significant increases in $R^2$ values, confirming its superior accuracy and predictive power.

Most notably, the HPA exhibited exceptional performance across both models. For the GRU model, it achieved the lowest RMSE (0.02) and MAE (0.013), and the highest $R^2$ (0.999). When applied to the CNN-R model, it maintained its high performance with an RMSE of 0.082, MAE of 0.082, and $R^2$ of 0.999. Compared to the best-performing traditional optimizer, GWO, the proposed method showed an RMSE improvement of approximately 93%, an MAE improvement of 94%, and a 7.7% improvement in $R^2$. These results highlight the potential of hybrid optimization techniques in enhancing machine learning model performance.

Despite showing encouraging results for grape moisture estimation, the suggested framework's applicability is currently restricted to numeric datasets. Furthermore, explainability issues should also be taken into account to see how specific features directly affect model decisions. Improving interpretability *via* feature importance analysis and

using HPA to classification tasks with image-based datasets will be the main goals of future research. Through this modification, the potential of HPA in feature selection optimization for various image classification applications—particularly in the fields of remote sensing and smart agriculture—will be investigated. We will be able to compare HPA's performance with state-of-the-art approaches in these future investigations by applying it in combination with deep learning models and other sophisticated feature selection techniques. We hope that this will provide us with a better grasp of how versatile and successful HPA is when working with different kinds of data, including both image-based and numeric information.

In conclusion, the CNN-R model, paired with advanced optimization techniques, offers a robust and highly accurate alternative to traditional GRU models. This combination is particularly advantageous for applications requiring precise and reliable predictions. Our findings provide a strong statistical basis for adopting CNN-R and hybrid optimization methods in future predictive analytics and machine learning research, setting a new benchmark for model accuracy and efficiency.

### Funding
The authors received no funding for this work. The APC is supported via funding from Prince Sattam bin Abdulaziz University, project number (PSAU/2025/R/1446). The funders had no role in study design, data collection and analysis, decision to publish, or preparation of the manuscript.

### Grant Disclosures
The following grant information was disclosed by the authors:
Prince Sattam bin Abdulaziz University: PSAU/2025/R/1446.

### Competing Interests
The authors declare that they have no competing interests.

### Author Contributions
- Ali Roman conceived and designed the experiments, performed the experiments, analyzed the data, performed the computation work, prepared figures and/or tables, authored or reviewed drafts of the article, and approved the final draft.
- Youssef Altherwy analyzed the data, authored or reviewed drafts of the article, and approved the final draft.
- Syed Rameez Naqvi conceived and designed the experiments, analyzed the data, prepared figures and/or tables, authored or reviewed drafts of the article, and approved the final draft.
- Anas Alsuhaibani analyzed the data, authored or reviewed drafts of the article, and approved the final draft.

## Data Availability

Raw data is available at Zenodo:

Roman, A., Altherwy, Y., Naqvi, S. R., & Alsuhaibani, A. (2025). Grapes Moisture [Data set]. Zenodo. https://doi.org/10.5281/zenodo.15637308.

## Supplemental Information

Supplemental information for this article can be found online at http://dx.doi.org/10.7717/peerj-cs.3142#supplemental-information.

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
