# Peer review of "A novel nature-inspired feature selection algorithm for efficient moisture estimation in fruits using RF-Sensed data"

_PeerJ Computer Science, doi:10.7717/peerj-cs.3142_

## Round 0.1 · original submission · Major Revisions

Dear Authors,

Your paper has been reviweed. It needs major revisions before being accepted for publication in PEERJ Computational Science. More precisely

1) You have uploaded a dataset file titled "cs-113079-A_-_Full_s11.csv" to the journal. However, this dataset does not align with the description provided in the manuscript. Additionally, the paper states, "The dataset captures the S-parameters (S11, S12, S21, S22) for each cluster across four distinct files," yet only one file appears to be included. Furthermore, the manuscript mentions that "they include 1001 frequencies and four angles," but it's unclear what the 1001 specifically refers to. Is this a typo (perhaps intended to be 1601, as used elsewhere), or does it refer to a subset? You must solve these issues.

2) The paper focuses on a nature-inspired feature selection algorithm. While Section 4.1, Estimation using GRU and Benchmark Techniques, presents some valuable results, it lacks essential implementation details. You must add them with the hyperparameters used for the benchmark algorithms. Furthermore, it would be beneficial to include an analysis of the selected features to better understand the effectiveness of the optimization approaches exploited in your study; specifically:
– Which features are selected by each algorithm?
– How many features are selected by each method, and why?

3) Comparing the proposed model's runtime efficiency against the baseline methods would strengthen the paper.

4) The Abstract of this article needs restructuring. It is not a place to prove the novelty of your work. I recommend that you be precise and clearly distinguish between the background, methods, results, and comparison with the state-of-the-art. The abstract in the current version is a mix, raw, and challenging to follow. Avoid using the above and below in the text to cite tables. "Provide a summary of some helpful uses for this technology below". Instead, call the Table.

**Language Note:** PeerJ staff have identified that the English language needs to be improved. When you prepare your next revision, please either (i) have a colleague who is proficient in English and familiar with the subject matter review your manuscript, or (ii) contact a professional editing service to review your manuscript. PeerJ can provide language editing services - you can contact us at [email protected] for pricing (be sure to provide your manuscript number and title). – PeerJ Staff

·

Basic reporting

The paper is titled as 'A Novel Nature-Inspired Feature Selection Algorithm for Efficient Moisture Estimation in Fruits Using RF-Sensed Data' and it is aimed to develop and validate a novel hybrid nature-inspired optimization technique for feature selection, improving the accuracy of fruit moisture estimation using RF-sensed data within a machine learning framework tailored for smart agriculture applications.

Experimental design

The methods are described with sufficient detail, allowing for a general understanding of the proposed approach. However, some sections could be streamlined or shortened to improve readability and focus.

Additionally, the manuscript lacks information on data preprocessing steps. This is an important aspect, especially when working with raw RF-sensed data, and should be included to ensure clarity and reproducibility.

The authors have uploaded a dataset file titled "cs-113079-A_-_Full_s11.csv" to the journal. However, this dataset does not appear to align with the description provided in the manuscript.

Firstly, the file lacks a target value, which is essential for supervised learning tasks like regression.

Additionally, the paper states that "The dataset captures the S-parameters (S11, S12, S21, S22) for each cluster across four distinct files," yet only one file appears to be included.

Furthermore, the manuscript mentions that "they include 1001 frequencies and four angles,"—but it's unclear what the 1001 specifically refers to. Is this a typo (perhaps intended to be 1601, as used elsewhere), or does it refer to a subset?

Could the authors clarify whether any parts of the dataset are missing, and if not, how the uploaded file represents the full data described in the paper?

Validity of the findings

The paper is written in a clear and professional manner, with a good level of English throughout. The language is generally fluent and does not hinder understanding of the content.

The paper makes use of recent and relevant literature, which is well-aligned with the topic. The cited works are appropriately chosen and support the context and motivation of the study effectively.

The authors have obtained valuable results that clearly demonstrate the advantages of the proposed algorithms. The performance improvements are well-supported by the experimental findings.

Additional comments

The topic is interesting and fits in the scope of the PeerJ Computer Science Journal.

However, I have several suggestions that could help improve the overall quality and clarity of the paper.

Major 1)
The authors have uploaded a dataset file titled "cs-113079-A_-_Full_s11.csv" to the journal. However, this dataset does not appear to align with the description provided in the manuscript.

Firstly, the file lacks a target value, which is essential for supervised learning tasks like regression.

Additionally, the paper states that "The dataset captures the S-parameters (S11, S12, S21, S22) for each cluster across four distinct files," yet only one file appears to be included.

Furthermore, the manuscript mentions that "they include 1001 frequencies and four angles,"—but it's unclear what the 1001 specifically refers to. Is this a typo (perhaps intended to be 1601, as used elsewhere), or does it refer to a subset?

Could the authors clarify whether any parts of the dataset are missing, and if not, how the uploaded file represents the full data described in the paper?

Major 2)
The paper focuses on a nature-inspired feature selection algorithm, and while Section 4.1, Estimation using GRU and Benchmark Techniques, presents some valuable results, it lacks important implementation details.

Specifically, I would like to see the hyperparameters used for the benchmark algorithms, as these play a crucial role in the performance of iteration-based optimization techniques.

Providing these details would enhance the reproducibility and credibility of the experimental results.

Major 3)
The paper focuses on feature selection using various optimization approaches.

However, to better understand the effectiveness of these approaches, it would be valuable to include an analysis of the selected features.

Specifically:
– Which features are selected by each algorithm?
– How many features are selected by each method, and why?

Such an analysis would provide deeper insight into the behavior of the optimization techniques and help validate the rationale behind the feature selection process.

Major 4)
It would strengthen the paper to include a comparison of the runtime efficiency of the proposed model against the baseline methods.

Since this work involves optimization-based feature selection, it implicitly introduces a multi-objective aspect, aiming not only to improve estimation accuracy but also to manage computational cost.

In many cases, improving one metric (e.g., accuracy) may come at the expense of another (e.g., runtime). Therefore, an analysis of trade-offs between performance and computational efficiency would provide a more comprehensive evaluation of the proposed approach.

Minor 1)
The title "Table 1. Related work regarding RF-sensed data" should be placed above the table, in accordance with standard formatting practices.

Additionally, in the same table, the citation "Oliveira et al. (2024)" should be formatted as "(Oliveira et al. (2024))" to maintain consistency with the citation style used for other entries.

Minor 2)
The sentence "Our estimation accuracy metrics include RMSE, MAE, and R², which report the best-case performance of 99.9%" is unclear, as these metrics represent different aspects of model performance—RMSE and MAE focus on error, while R² indicates goodness of fit.

It is not specified which metric the 99.9% refers to.

Please clarify which specific metric achieved this value.

Minor 3)
In the Abstract, the authors state: "The proposed technique follows a unique method of hybridizing multiple nature-inspired algorithms, where the general framework is adopted from one, while its position update equations are inspired by the other existing algorithms."

However, it is unclear what "one" and "other existing algorithms" specifically refer to.

For clarity, please explicitly mention which algorithm provides the general framework and which ones contribute to the position update equations.

Minor 4)

The sentence "The reason for choosing LSTM, BI-LSTM, and GRU was that the dataset discussed in 2.2 is sequential by nature." should be updated for clarity and consistency.

Specifically, "2.2" should be written as "section 2.2" to maintain proper academic referencing style.

Additionally, note the typo in "chosing", which should be corrected to "choosing."

Minor 5)
The inclusion of Equations 1 through 6 may not be necessary, as these formulations are not directly utilized or analyzed within the scope of the paper.

They appear to be standard operations that are implemented through existing libraries and are embedded within the underlying algorithms.

Therefore, emphasizing their detailed mathematical formulations does not contribute significantly to the manuscript and could be removed to improve focus and conciseness.

Minor 6)

Although the paper has good English. I have some corrections as follows:
CNN consist of multiple layers, each with a specific role: -- CNN consists of multiple layers, each with a specific role:
are usually considered most suited models --are usually considered the most suited models
models for a similar application on the same dataset -- the, a!
As our previous study suggests better performance in terms--as our previous study suggests better performance in terms

Reviewer 2 ·

Basic reporting

The paper investigates a framework for estimating moisture in grapes using RF sensor data, incorporating a combination of Gated Recurrent Units (GRU) and Convolutional Neural Networks (CNN) to propose CNN-R. The structure is complete, and the logic is clear. However, the paper presents the following issues:

Experimental design

1. The literature review should be expanded to include an analysis of methods based on transformers and mamba (such as RSMamba, DynamicVis, etc.).
2. The statement of related work should more effectively highlight the research challenges and motivations.
3. The figures in the paper should be replaced with more comprehensible schematic diagrams rather than flowcharts.
4. The experiments are insufficient and should include comparisons with other methods.

Validity of the findings

-

Reviewer 3 ·

Basic reporting

Overall, the article looks good. However, I suggest that authors address the following comments:

1. Abstract of this article needs restructuring. It is not a place to prove your novelty of work. I recommend that you be precise and bring a clear distinction between the background, methods, results, and comparison with the state-of-the-art. The abstract in the current version is a mix, raw, and difficult to follow.

2. Avoid using above and below in text for citing tables. "Provide a summary of some helpful uses for this technology below". Instead, call the Table.

3. One of the main contributions of this article is proposing a mutation-enhanced hybrid optimization algorithm named HPA (Hybrid Predator Algorithm), for superior feature selection. I went through Table 4, where authors defend for computational costs. I believe the steps involved in the HPA algorithm will result in high computational costs, especially with high-dimensional datasets. I just want authors to bring clarity here while making inferences in Table 4.

4. I would appreciate it if the authors could add details on how convergence issues of the model were addressed for your chosen dataset.

5. I suggest that authors add a table for comparison of their current work results with their previous work, along with the existing works from the state-of-the-art. In Fig.1, you are giving a comparison between your existing approach with the current one, raising expectations in the readers. You must prove this claim before ending the article in terms of results or outcomes.

4.

Experimental design

Experimental design is alright. Only a few comments need to be addressed as given in Section 1.

Validity of the findings

This section required a change as per the comments given in section 1.

---

## Round 0.2 · Minor Revisions

Dear Author,
Your paper has been revised. It needs minor revisions before being accepted for publication in PEERJ Computer Science. More precisely:

1) To further improve the visual quality and consistency of the submission, it is recommended that you revisit Table 11, Figure 6, and Figure 7. Please ensure that all numerical values are displayed with a consistent number of decimal places.

2) You should consider removing any unnecessary blank space in the upper sections of Figures 6 and 7 to enhance their visual presentation.

·

Basic reporting

The manuscript generally meets the journal’s expectations in terms of clarity, structure, and presentation. The English is mostly clear, understandable and professional. The Introduction sets the context well and outlines the motivation clearly. The structure aligns with PeerJ standards and disciplinary norms, with a logical flow. Minor deviations appear intentional and aid clarity. 8It is explained in the ongoing sections)

Experimental design

The article titled as "A novel nature-inspired feature selection algorithm
for efficient moisture estimation in fruits using RFSensed data" seems to fall well within the PeerJ Computer Science journal’s scope and fits the article type appropriately. The results appears technically sound and ethically handled. Methods are explained fairly clearly, and giving related details. The part on data preprocessing is mostly sufficient.

Validity of the findings

The results are valuable and support the aim of the paper.

Additional comments

According to the previous round of reviews, the authors have made the necessary corrections in response to the comments provided.

The current version of the manuscript is acceptable in its present form.

Minor Suggestion:
To further improve the visual quality and consistency of the submission, it is recommended that the authors revisit Table 11, Figure 6, and Figure 7. Please ensure that all numerical values are displayed with a consistent number of decimal places. Additionally, consider removing any unnecessary blank space in the upper sections of Figures 6 and 7 to enhance their visual presentation.

---

## Round 0.3 · Minor Revisions

Dear Author,
Your paper has been revised. It needs minor revisions before being accepted for publication in PEERJ Computer Science. More precisely:

1) You should better highlight the innovations of the proposed method and elaborate on how it differs from existing approaches for the same task.

·

Basic reporting

This is the third round of review, and the authors have successfully implemented the minor revisions requested. As the changes were minimal and have been adequately addressed, there are no additional detailed comments at this section.

Experimental design

This is the third round of review, and the authors have successfully implemented the minor revisions requested. As the changes were minimal and have been adequately addressed, there are no additional detailed comments at this section.

Validity of the findings

This is the third round of review, and the authors have successfully implemented the minor revisions requested. As the changes were minimal and have been adequately addressed, there are no additional detailed comments at this section.

Additional comments

The authors have addressed the previously raised concerns and made the necessary corrections throughout the manuscript. The revisions have improved the clarity and quality of the work, and all major issues appear to be resolved. Therefore, I believe the paper is now suitable for publication and can be accepted as is.

Reviewer 2 ·

Basic reporting

The authors propose a hybrid nature-inspired feature selection algorithm that incorporates update mechanisms from the Grey Wolf Optimizer (GWO), Artificial Bee Colony algorithm (ABC), and Bat Algorithm (BA). This approach enables the optimization of machine learning models for the accurate estimation of grape water content using radio frequency (RF) sensing data. The manuscript is clearly written, logically structured, and supported by comprehensive experiments. The paper may be accepted after minor revisions, particularly to better highlight the innovations of the proposed method and to elaborate on how it differs from existing approaches for the same task.

Experimental design

N/A

Validity of the findings

N/A

Additional comments

N/A

---

## Round 0.4 · accepted · Accept

Dear Author,

Your paper has been revised. It has been accepted for publication in PEERJ Computer Science. Thank you for your fine contribution.

In particular, the Section Editor noted that:

> If headings are to be numbered, the introduction section should also be included. Alternatively, if required by the journal format, all heading numbers should be removed, and the heading hierarchy should be restructured accordingly.

Reviewer 2 ·

Basic reporting

The paper can be accepted.

Experimental design

The paper can be accepted.

Validity of the findings

The paper can be accepted.

Reviewer 3 ·

Basic reporting

All of my comments are addressed and revised version looks much improved.

Experimental design

NA

Validity of the findings

NA

Additional comments

NA